

**Landslide Hazard Microzonation Using a Hybrid Integrated Approach to Reduce**
**Disaster Risk: A Case Study of Jecheon, South Korea**
**Jae-Joon Lee[1], Manik Das Adhikari[2], Moon-Soo Song[2], Sang-Guk Yum[2]\***
[1] Department of Fire Safety Engineering, Jeonju University, Jeonju, Jeollabuk-do, Republic
of Korea
[2] Department of Civil and Environmental Engineering, Gangneung-Wonju National
University, Gangneung, Gangwon-do, Republic of Korea
*Corresponding author: Sang-Guk Yum; skyeom0401@gwnu.ac.kr*
**Abstract**
Effective landslide prevention and mitigation necessitate the development of reliable landslide
susceptibility map. However, previous studies have primarily focused on assessing the overall
performance of predicted susceptibility rather than examining the spatial characteristics of the
predicted Landslide Susceptibility Index (LSI). This study aims to evaluate the efficacy of
predicted LSIs derived from widely used statistical models while considering the spatial
characteristics of landslides. To achieve this goal, four commonly used LSI models, namely
frequency ratio (FR), certainty factor (CF), logistic regression (LR), and information value
(IV), were utilized to map landslide susceptibility in Jecheon, South Korea. The models were
developed using 112 landslide inventories and taking into account topography, hydrogeology,
soils, forests, and lithological heterogeneities. Subsequently, the predicted LSIs were compared
with the 1D topography profiles of recent landslide events delineated from the high-resolution
aerial and drone imagery. The distribution of anticipated LSIs along the landslide source area
to the landslide runout and deposit zones was found to be inconsistent with the landslide
characteristics. Nevertheless, the overall accuracy of the FR, IV, CF, and LR models
demonstrated the strong predictive capabilities of these models. To address this spatial
inconsistency issue, we proposed a hybrid integrated approach to achieve higher accuracy than
the individual LSI models. Subsequently, a landslide hazard microzonation map was prepared
and validated based on the in-situ observations and inventory data. It was observed that 94.6%
of landslide inventory occurrences fell within severe to high-hazard zones. Precision results,
such as an area under the curve of 0.906, mean square error of 0.25, mean absolute error of



0.08, root mean square error of 0.28, and a precision of 88.3%, suggest that the hybrid
integrated approach is more useful for landcover planning and mitigating landslide-induced
disaster risks compare to individual LSI models.
**Keywords:** Landslide susceptibility, Logistic regression, Certainty factor, Frequency ratio,
Information value, Hybrid Integrated approach, Accuracy
**1. Introduction**
Landslides are geologic events in which large amounts of soil and rock detach from the ground
and move downslope, potentially causing damage and destruction in their path. The damage
caused by landslides is critical on a global scale. During the last century, landslides have
resulted in thousands of fatalities and billions of dollars in property damage (Chen and Chen,
2020; Lee et al., 2017). Due to high-intensity rainfall and a changing climate, landslides have
become more frequent in recent years (Lee et al., 2018). Consequently, numerous researchers
(Dash et al., 2022; Mandal et al., 2021; Pham et al., 2020; Shano et al., 2020; Zhou et al., 2018;
Aditian et al., 2018; Ghosh and Bhattacharya, 2010) have pursued work to predict and prevent
landslide hazards using various methods.

The Korean peninsula's geological fragility, mountainous topography, and frequent

typhoon-induced heavy rainfall make it prone to deadly landslides. In recent decades, these
landslides have caused significant loss of life and property damage. Lee and Winter (2019)
reported that more than 1728 fatalities occurred between 1970 and 2017 in the Korean
peninsula and an annual financial loss of about US$ 500M due to landslides. In South Korea,
a large population resides in landslide susceptible regions (Lee et al., 2002). In addition, it is
anticipated that climate change, urbanization, and timber harvesting will increase the frequency
and severity of landslide-induced damages (Park et al., 2019). Therefore, a reliable landslide
hazard potential mapping is crucial in understanding the fundamental concepts of risk
assessment and its impact.

61       The landslide hazard microzonation assesses the potential for natural slope instability

in a given area (Peethambaran and Leshchinsky, 2023). It involves evaluating the physical
factors that can lead to landslides and creating a map to show the relative likelihood of such an
event occurring in a given area. The resulting landslide susceptibility maps indicate the regions
most likely to experience landslides (Guzzetti et al., 1999). In the past two decades, several
statistical and machine-learning models for landslide susceptibility analysis have been




suggested, presuming that landslides trigger in a similar environment to prior landslides (Wei
et al., 2023; Reichenbach et al., 2018; Park et al., 2013; Lee and Pradhan, 2007; Lee et al.,
2002). Although numerous techniques have been put forth to create GIS-based landslide
susceptibility maps, there still needs to be an agreement on the best practices (Aditian et al.,
2018). Most quantitative methods considered past landslides to determine the ranks and weight
of each factor attribute based on their spatial association. Subsequently, several quantitative
methods, including frequency ratios, Shannon entropy, certainty factor, logistic regression,
information value, weights of evidence, support vector machine, neural networks, random
forest, and hybrid models, are frequently applied to landslide potential mapping (e.g., Park et
al., 2023; Dash et al., 2022; Mandal et al., 2021; Pham et al., 2020; Aditian et al., 2018; Riaz
et al., 2018; Zêzere et al., 2017; Shahabi and Hashim, 2015; Wang et al., 2015). The benefits
and drawbacks of several probabilistic and statistical approaches were recently reviewed by
Merghadi et al. (2020) and Shano et al. (2020). Even though there were numerous studies on
landslide susceptibility, no single approach is suitable for all cases. As a result, to determine
landslide susceptibility in a given area, the best model must be chosen based on the
landslide's characteristics and the accessibility of inventory data (Zhu et al., 2018).
Consequently, it is still crucial to calculate the effectiveness of various models for particular
landslide susceptibility procedures. In addition, model integration provides another opportunity
to improve model accuracy by combining different models on the GIS platform.
A landslide susceptibility index typically indicates areas that are more prone to
landslides based on various factors and parameters. Thus, previous studies have primarily
focused on assessing the overall performance of predicted susceptibility rather than examining
the spatial characteristics of the predicted Landslide Susceptibility Index (LSI). The overall
accuracy of widely accepted models may produce acceptable LSIs in terms of AUC, MAE, and
RMSE, but they may not always be comparable with the landslide characteristics. For example,
the landslide source area is the region at the top of a slide where the slope begins to fail, and
the movement of the soil, rock, and other material begins (Lee et al., 2002). Thus, the predicted
LSI value should be higher in the source and crown zones. On the other hand, the landslide
deposit area is the region at the bottom of the slide, where the material from the source area
ends up after it has moved downslope and poses further vulnerability. Consequently, the LSI
value in the landslide-deposited area must be lower than the source area. Therefore, the
landslide characteristics along with the AUC, MAE and RMSE, should be used to validate the
predicted LSI values (Lee et al., 2002). However, most landslide studies consider the overall




model performance (i.e., AUC) and ignore the spatial inconsistency phenomenon. Therefore,
the main novelties of this study include (a) the development of landslide susceptibility (LSI)
maps by comparing and analyzing different statistical models commonly used for assessing
LSI, (b) evaluating spatial characteristics of the predicted landslide susceptibility indexes to
study previously overlooked accuracy criteria, (c) proposed a hybrid integrated approach to
achieve higher accuracy than the individual LSI models, and (d) prepared a reliable landslide
hazard microzonation map to mitigate landslide-induced disaster risks appropriately.
**2. Study area**
The mountainous region of South Korea is prone to rainfall-induced landslides, causing
fatalities and extensive damage to roads, bridges, and settlements. Over 70% of the Korean
peninsula has steep mountain slopes (>30°) (Lee et al., 2022a; Lee et al., 2015). Rainfall
accompanied by occasionally severe typhoons has adversely affected this region (Lee et al.,
2022b). In contrast, shallow landslides often occur throughout the rainy season (June to
September) under different geological conditions (Kim et al., 2021). According to the Korea
Forestry Service's analysis of landslide extent, the annual average landslide area rapidly
increased from 231 hectares in the 1980s to 713 hectares in the 2000s (Lee et al., 2018). The
present work focuses on the Jecheon-si region (36°48′47"-37°16′ 15"N, 127°55′19"-128°20′
E), situated in the northern part of North Chungcheong Province and covers an area of ~
884.3372 km$^2$ (Fig. 1). This region is surrounded by mountains, lake (Cheongpung Lake) and
reservoirs. Geologically, the region is situated in the southwestern part of the Gyeonggi Massif,
which is composed of the metamorphic basement and sedimentary strata (Seo et al., 2011). The
surface geology is mainly covered by sandy mudstone, mudstone, quaternary loess strata
outcrop and sandstone (Jung and Kang, 2014; Kihm et al., 2000). The topography of Jecheon-
si is mainly composed of mountains with the highest altitude of 1157m and the lowest elevation
of 117m. The city exhibits dispersed high-density settlements (Fig. 1b), which makes it very
densely populated in some areas. The original terrain was altered during the urbanization due
to engineering activities (i.e., road construction), resulting in slope deformation and instability
in this region.





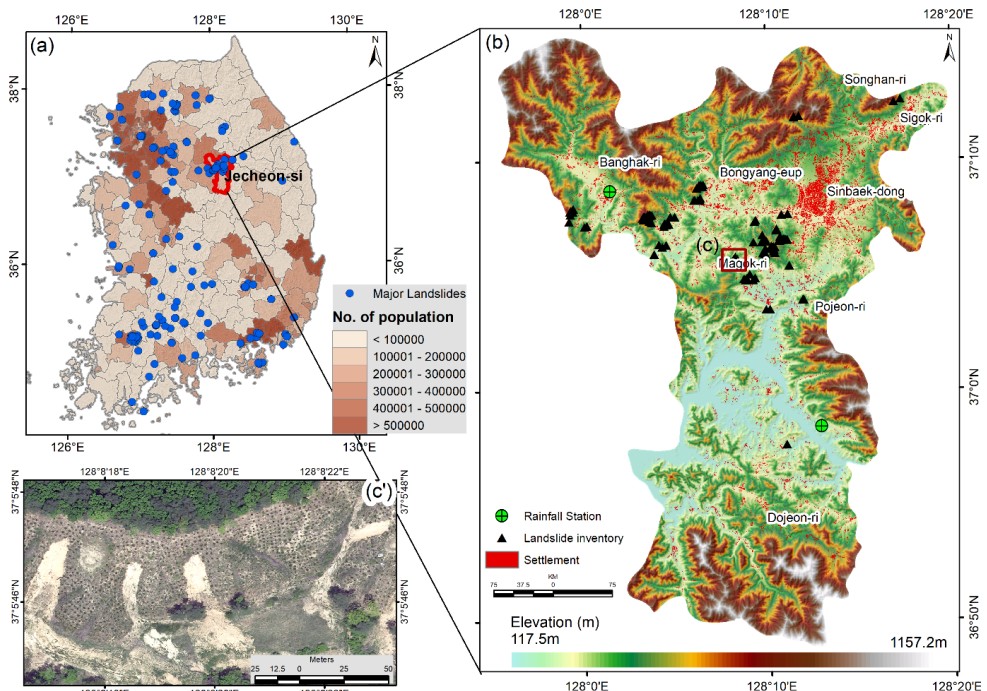

**Fig. 1** Location of the study area (Jecheon-si): **a** major landslide distribution of South Korea during 2007-2020 reported by Lee et al. (2022b), **b** updated landslide inventory of Jecheon-si region, and **c** typical landslide detected from the aerial photo (aerial image acquisition 2021, http://map.ngii.go.kr/ms/map/ ).

Jecheon region receives 1,360.9mm of rainfall annually. The highest rainfall in a single month was 265.2mm in August, and the lowest amount recorded in a single month was 26.9mm in February. Rain can cause various changes in the soil, including increased saturation and decreased stability. Increased saturation can cause the soil to become more prone to slippage and movement, while decreased stability can lead to soil becoming more susceptible to erosion (Pradhan and Kim, 2016). Additionally, heavy rainfall can trigger landslides by loosening rocks and debris on steep slopes (Pradhan and Kim, 2014). Furthermore, intense rainstorms can also cause streams and rivers to swell rapidly, leading to increased erosion and landslide risk. The region is likely to have an increased risk of landslides when there is high-intensity rainfall in a short period, especially during August and September when the most rainfall is recorded, as observed in 2020 (Lee et al., 2022b). In the present study, the short-term rainfall characteristics were analyzed in the past 28 days before the recorded landslide event on 3$^{rd}$ August 2020, at



Jung-myeon, and 2ⁿᵈ August 2020, at Wonbak-ri, in the Jecheon-si region as depicted in Fig.
2. We obtained rainfall data from the automatic weather station (AWS) (https://data.kma.go.kr/
) located in Jecheon-si on a daily basis (Fig. 1b). At both sites, the accumulated rainfall was
536mm and 514.5mm, respectively, before the recorded landslide event.

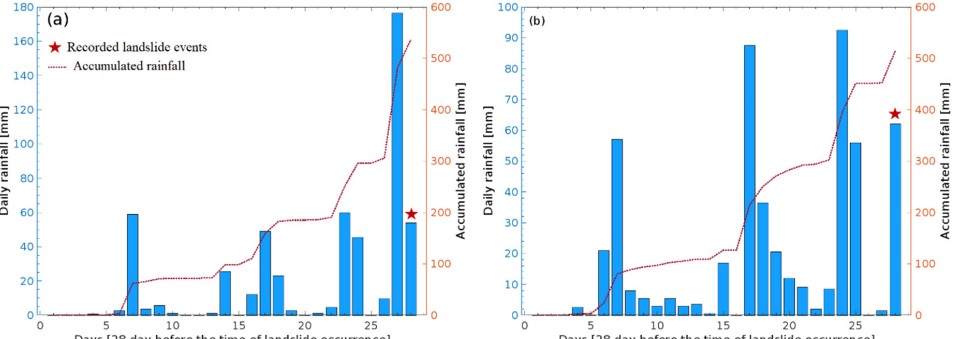


**Fig. 2** Rainfall characteristics in the past 28 days before the recorded landslide event on **a** 3ʳᵈ
August 2020, at Jung-myeon, and **b** 2ⁿᵈ August 2020, at Wonbak-ri, in the Jecheon-si
region (Data Source: https://data.kma.go.kr/ ).

**3. Data and methods**
Landslide susceptibility analysis was performed using the FR, CF, IV, and LR models based
on spatial and non-spatial data. This study used the following steps to analyze landslide
susceptibility: (1) creating spatial data on landslide predisposing factors and a detailed
landslide inventory database, (2) the relationship between landslide predisposing factors and
inventory analyzed using the FR model, (3) the FR, IV, CF and LR models were performed
using MATLAB and ArcGIS software, (4) comprising of predicted LSI values with the
topographic and landslide characteristics of a few past landslide events, and (5) integration of
four LSI models (i.e., FR, IV, CF and LR) on the GIS platform and evaluated the accuracy of
integrated and individual models using R-Index, RMSE, MAE, MSE and ROC. The workflow
of this study is shown in Fig. 3. The detailed data and methods used in the present study were
discussed in sections 3.1 and 3.2.



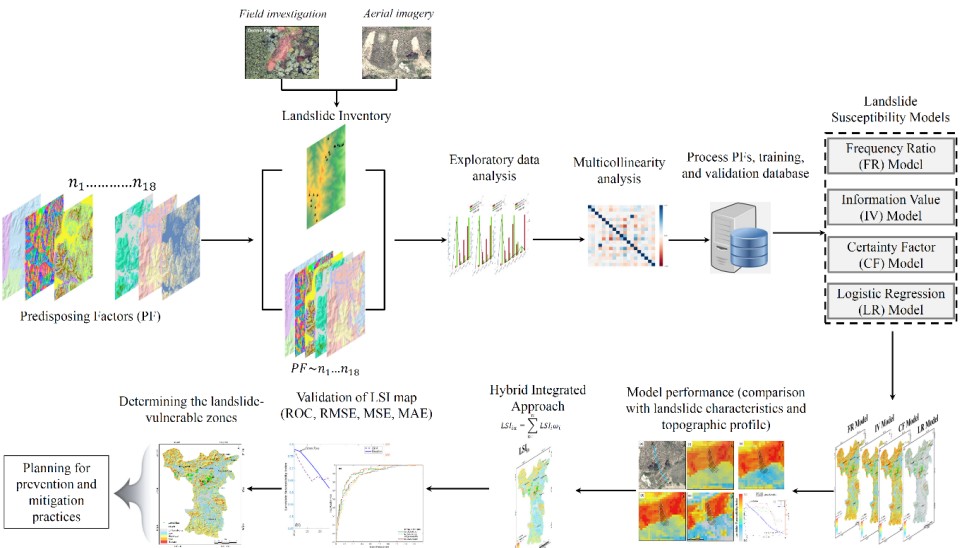


**Fig. 3** Workflow of the analytical framework and detailed steps of landslide hazard
microzonation mapping proposed in the present study.


## 3.1. Landslide inventory and predisposing factors

### 3.1.1 Preparation of landslide inventory database

A landslide inventory database presents the location and characteristics of prior
landslides. Thus, the inventory map offers valuable information regarding the spatial and
temporal distribution of existing slope failures and the potential of future slides (Choi et al.,
2012). Conversely, creating an inventory database is essential for evaluating the accuracy
statistics of landslide potential maps (Park et al., 2019). In order to create a landslide inventory
database, various methods can be employed, including field assessments, satellite imagery, and
aerial photography. In the present study, the inventory database was created using aerial
photographs (available at http://map.ngii.go.kr/ms/map/), historical Google Earth imagery, and
field investigation. The best way to get an image of a landslide inventory is to use Google Earth
(Kadavi et al., 2018; Van Den Eeckhaut et al., 2012). Google Earth is a powerful tool allowing
users to view satellite imagery and aerial photographs of a location on a multi-temporal scale.
In addition, the boundaries of the landslides were mapped using dronographs and aerial
imageries. The landslide boundary is presented using polygon data, and point data indicate the
landslide crown zones, source areas, runout and depositional areas, as depicted in Fig. 4.
Subsequently, we detected 112 landslides spread over the Jecheon-si region, as shown in Fig.





1b. Further, the region situated 130m from the landslide origin was considered a stable region,
as the maximum runout length of a landslide in this region is approximately 130m.
Consequently, the non-hazardous cells within these stable zones hold significant importance in
any probabilistic model (Giarola et al., 2024).

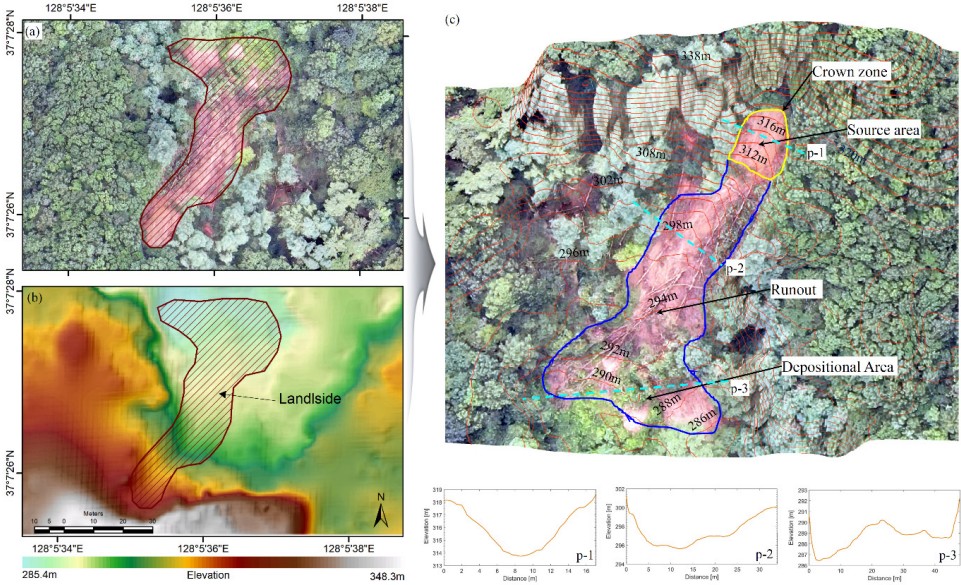


**Fig. 4 a** dronograph of a shallow landslide captured during the field investigation, **b** high-
resolution DEM constructed from drone orthophotos, and **c** landslide source and runout
area marked in yellow and blue color, respectively. The subplots (**p-1** to **p-3**) represent
the crossectional elevation profile of the source, runout and accumulation areas (marked
in Fig. 4c).

### 3.1.2. Predisposing factors

Rainfall-induced landslides in mountainous regions are quite frequent. In order to
accurately assess landslide hazards, a deep understanding of the landslide characteristics and
its mechanics is often necessary. Several factors influence the initiation of debris flows,
including topography, hydrology, lithology, soil and forest (Table 1). The relevant topographic
predisposing factors used for the landslide susceptibility model are the slope, aspect,
topographic position index (TPI), convergence index (CI), topographic roughness index (TRI),
plan curvature, profile curvature, and landforms. The significant hydrological predisposing
factors include slope length (SL), stream power index (SPI) and topographic wetness index
(TWI) that characterize debris material's concentration, dispersion, and balance on slopes. In



addition, soil (i.e., soil texture, soil thickness), lithology (i.e., surface lithology, average shear-
wave velocity), and timber factors (i.e., timber density, diameter and ages) influence the
geographic distribution of landslide events. The topographic and hydrologic predisposing
factors were generated from a high-resolution DEM (5 × 5 m grid) using ArcGIS, QGIS and
SAGA GIS software. The soil, lithology, and timber factors were extracted from the digital
soil, geology, and forest database. On the other hand, the velocity-slope model proposed by
Wald and Allen (Wald and Allen, 2007) was used to generate the subsurface properties of rock
and soil. Subsequently, we considered 18 predisposing factors for landslide susceptibility
analysis of the Jecheon-si region.

**Table 1** Description of landslide predisposing factors.

| Data | Factors | Data Types | Scale | Sources |
|---|---|---|---|---|
| Topographic Factors | Slope | GRID | 1:5000 | National Geographic Information Institute (NGII) |
| | Aspect | | | |
| | plan curvature | | | |
| | profile curvature | | | |
| | topographic position index (TPI) | | | |
| | convergence index (CI) | | | |
| | topographic roughness index (TRI) | | | |
| | landforms | | | |
| Hydrological Factors | slope length (SL) | GRID | 1:5000 | NGII |
| | topographic wetness index (TWI) | | | |
| | stream power index (SPI) | | | |
| Forest Factors | Timber Density | Polygon | 1:25000 | The forest map produced by Korea Forest Service (KFS) |
| | Timber Diameter | | | |
| | Timber Age | | | |
| Soil Factors | Soil Types | Polygon | 1:25000 | The detailed soil map produced by the Rural Development Administration (RDA) |
| | Soil Thickness | | | |
| Surface and sub-subsurface geology Factors | Surface Geology | Polygon | 1:50000 | Korean Institute of Geoscience and Mineral Resources (KIGAM) |
| | Time average shear-wave velocity ($V_s^{30}$) | GRID | 1:5000 | NGII |
| Rainfall | Daily Rainfall data | Observations | - | KMA (https://data.kma.go.kr/c) |


*3.1.2.1 Topographic factors*
Landform and topography are crucial in the formation of landslides. The Korean
National Geographic Information Institute (NGII) provided a high-resolution (5m×5m) digital
elevation model (DEM). After that, the pertinent topographic predisposing factors, i.e., slope,



aspect, TPI, CI, plan curvature, profile curvature, TRI, and landforms, were derived from DEM for susceptibility modeling.

The slope is a crucial indicator of the landslide process and is used in almost all landslide susceptibility studies (Fadhillah et al., 2022). The slope's gravitational potential energy is greater at higher elevations than at lower elevations. Generally, landslides occur more frequently on steeper slopes (> 25°) than on flatter slopes (Lee and Min, 2001). Consequently, the slope angle affects the weak rock and soil strata on the slope in terms of their strength and movement rate. Slope stability generally decreases with increasing slope angle. The slope angle of the Jecheon-si region ranges from 0 to 76°, as shown in Fig. 5a.

The topographic aspect controls the movement of water flow, vegetation and sun radiation, which influence landslide behaviors and types (Panahi et al., 2020). It represents the highest downhill slope. Additionally, slopes that face north or east are more likely to experience landslides due to increased exposure to moisture from prevailing winds. The topographic aspect value was classified into nine categories, as presented in Fig. 5b.

Slope instability is influenced by curvature, representing slope variations over a curve's tiny arcs. The plan and profile curvature of the Jecheon-si region ranges from -50.57 to 26.47 and -47.26 to 50.98, respectively, as shown in Figs. 5c & d. In general, convex surfaces are typically represented by positive curvature values, while concave surfaces are indicated by negative curvature values (Lee and Min, 2001). Negative curvature values have a higher likelihood of triggering landslides. On the other hand, Profile curvature describes the direction of the maximum slope and affects flow (convergence and divergence) across the surface (Oh and Lee, 2017).

The convergence index (CI) is an essential topographic predisposing factor for landslide susceptibility analysis. The CI of the Jecheon-si region ranges from -24.58 to 22.57, as shown in Fig. 5(e). The positive values of the convergence index represent ridges, whereas negative ones represent regional depressions (Petschko et al., 2014). In addition, the secondary geo-morphometric parameters, known as the terrain roughness index (TRI), characterize the local relief (Saha et al., 2021). The TRI determines the local terrain's roughness, which influences topographic and hydrological processes critical for developing landslides. Furthermore, the TRI also describes the state of drainage flow in a given area, which helps identify potential landslides. The TRI of the Jecheon-si region ranges from 0 to 14, as depicted in Fig. 5f.

The TPI is a numerical measure of a given location's relative elevation compared to its surrounding area. It represents the terrain's erosion/accumulation capacity (Park et al., 2019).





The negative TPI values signified lower elevated features than the surrounding features, and
values close to zero are represented as flat areas. In comparison, the positive values indicated
typically higher elevated features (Kadavi et al., 2019). The spatial value of TPI varies between
-25.7 and 29.6, as depicted in Fig. 5g.

The morphological setting of the area, crucial in regulating and expressing

morphodynamic activity, is directly distinguished by the landform classification (Martinello et
al., 2022). Therefore, the landform classification was performed using high-resolution DEM
data. The entire region was classified into ten landform classes, i.e., high ridges, local ridges,
mid-slope drainages, mid-slope ridges, plains, streams, upland drainages, open slopes, upper
slopes and valleys (Fig. 5h).

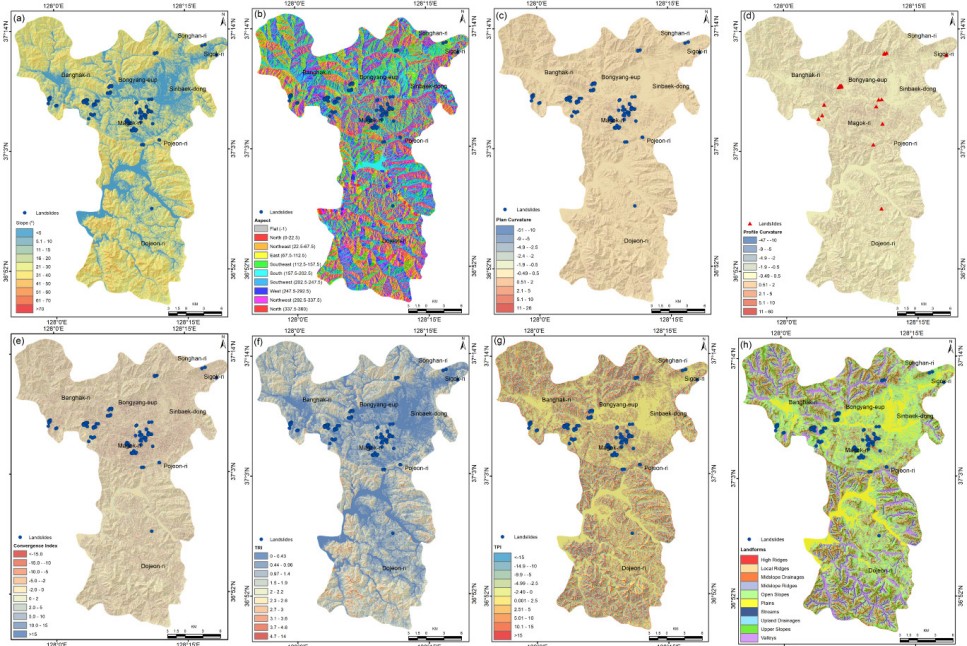


**Fig. 5** Topographical predisposing factors of Jecheon-si region.
*3.1.2.2 Hydrological factors*

The SL, TWI, and SPI are hydrological factors used for the susceptibility analysis.

Slope length (SL) is the most critical hydrologic predisposing factor that significantly impacts
landslide likelihood. Generally, the distance from the slope's crest to its toe is known as the
slope length. The SL is intimately associated with the development of landslides due to the
downhill movements of slope materials increasing with the slope length. Thus, the size of the



debris grows with a longer slope length (Qiu et al., 2018). The SL of the Jecheon-si region
varies from 0.0 to 423.5, as depicted in Fig. 6a.

The TWI evaluates the topographic effects of hydrological processes by considering

slope and flow direction (Panahi et al., 2020). It affects landslide occurrences in mountainous
regions (Sameen et al., 2020). The spatial value of TWI varies between 1.98 and 23.3, as shown
in Fig. 6b.

The SPI is characterized as the motion of granular material caused by gravity and the

erosive power of flowing water (Sameen et al., 2020). It describes the likelihood of flow erosion
at a specific point on the topographic surface. The spatial distribution of SPI of the Jecheon-si
region varies between 0.68 and 25.41, as depicted in Fig. 6c. Higher SL, SPI, and TWI values
indicate greater landslide susceptibility (Sameen et al., 2020; Qiu et al., 2018).
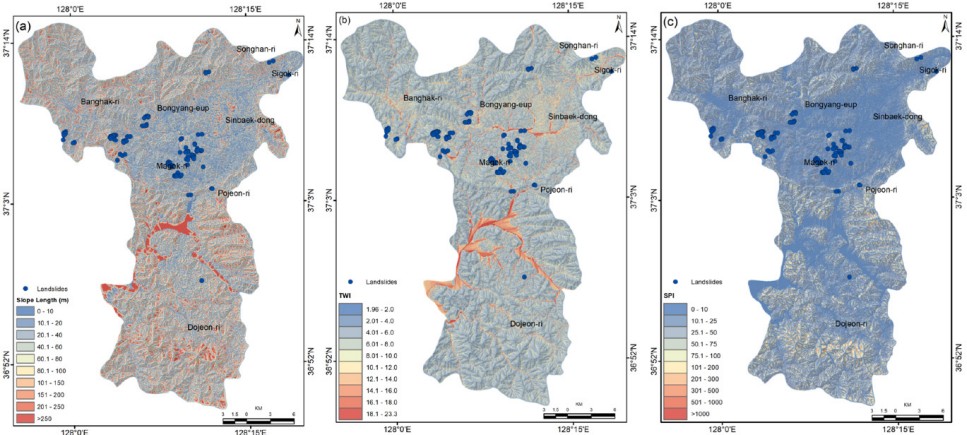
**Fig. 6** Hydrological predisposing factors of Jecheon-si region: **a** slope length (SL), **b**

topographic wetness index (TWI), and **c** stream power index (SPI).


*3.1.2.3 Lithological factors*

Outcropping lithology is the most important predisposing factor for landslide

evaluation, which is regarded as a good representation of rocks' physical-mechanical
characteristics (Lee and Min, 2001). The type and shape of mass movement are mostly
controlled by surface geology (Petschko et al., 2014). Furthermore, the occurrence of landslides
and their mechanisms could be predicted directly by the geological structure and subsurface
rock and soil properties (Panahi et al., 2020). Lithologically, landslides commonly occur in
weak rock layers and soft structural planes (Lee and Min, 2001). The outcropping lithology



layers of the Jecheon-si region are shown in Fig. 7a. Most of the study region was covered by
granitic rocks (syenite, hornblende, gabbro, diorite, etc.) and metamorphic rocks (phyllite,
gneiss, quartzites, etc.) (Jung et al., 2014).

Further, to understand the subsurface properties of rock and soil, a geotechnical site

classification in compliance with the NEHRP nomenclature using effective shear-wave
velocity ($V_s^{30}$) based on the topographic gradient (Wald and Allen, 2007) was performed.
Figure 7b depicts the seismic site classification of the Jecheon-si region, which exhibits the
following site classes: B+ ($V_s^{30}$: >760 m/s), C ($V_s^{30}$: 360-760 m/s), D ($V_s^{30}$:180-360 m/s) and
E ($V_s^{30}$: <180 m/s). The shear-wave velocity reflects the strength and impedance contrast
between the various soil/rock layers (Wald and Allen, 2007). Site classes E and D are
associated with stiff soils and the soft clay layer, whereas site class B is associated with hard
and compact rock. Thus, the average shear-wave velocity is extremely significant in identifying
the potential landslide zones (Abd El-Raouf et al., 2021).

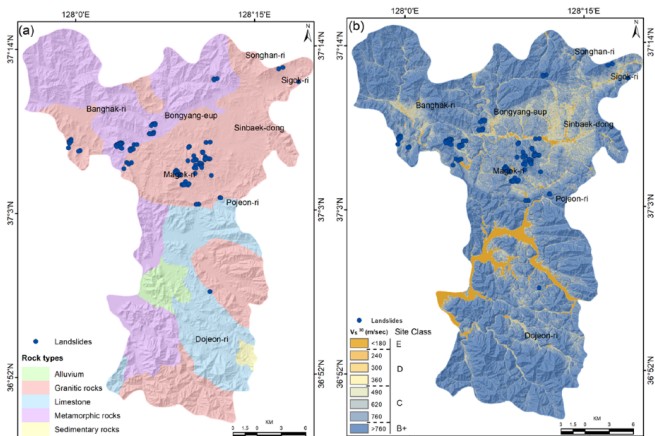


**Fig. 7** Lithological predisposing factors of the Jecheon-si region: **a** surface lithology, and **b**
average shear-wave velocity ($V_s^{30}$).


*3.1.2.4 Soil factors*

A soil's permeability and porosity relate to the soil material, influencing the fluid flow

of the region (Lee and Min, 2001). On the other hand, the amount of runoff and the soil's
capacity to absorb water is influenced by the soil's thickness (Sameen et al., 2010). Thus, the
thickness and texture of the soil are crucial factors in landslide susceptibility assessments. The
soil parameters have been extracted from a soil map developed by the Korea Rural
Development Administration (KRDA) and used as landslide predisposing factors. Soil texture





of the study region included eight classes: sandy clay loam, sandy loam, loam, silt loam, silty
clay, silty clay loam, and silty, as depicted in Fig. 8a. Various studies have shown that sandy
and clayey soil is more erosion-resistant than soil with a high silt concentration (Fonseca et al.,
2017). On the contrary, soil depth influences the shear stress and shear strength of rock and
soil on a slope (Pradhan and Kim, 2014). The depth of the soil also influences the volume of
the landslide. The soil thickness of the Jecheon-si region was divided into four classes: very
shallow (<20cm), shallow (20–50cm), moderate (50–100cm), and deep (>100cm), as shown in
Fig. 8b.

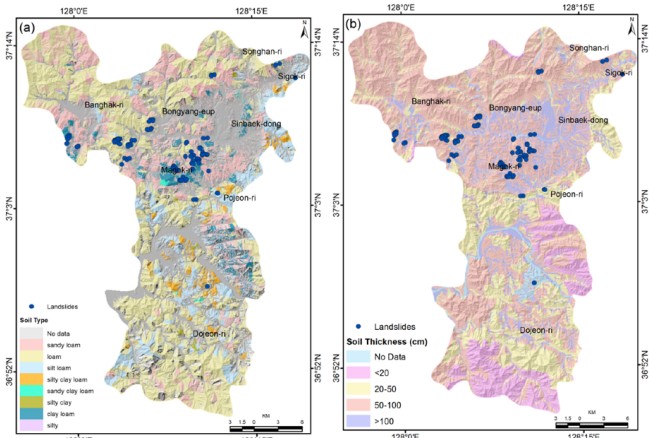


**Fig. 8** Soil predisposing factors of Jecheon-si region: **a** soil types and **b** soil thickness.

*3.1.2.5 Forest factors*

Forest characteristics such as density, diameter, and age are important attributes for

landslide susceptibility modeling (Fadhillah et al., 2022). The timber parameters were extracted
from the Korea Forest Research Institute and used as landslide predisposing factors. The
strength of soil-root connections significantly influences landslides (Kadavi et al., 2019). Thus,
forests with medium to large soil-holding capacities have the lowest probability of landslides
than non-forest regions (Lee et al., 2004).

The timber diameter was classified into three sizes: large (>30 cm), medium (18-30

cm), and small (<18 cm), as shown in Fig. 9a. The root system's density, which supports and
stabilizes the soil, is correlated with the density of the forest (Sameen et al., 2020). For example,
an area with dense vegetation can provide stability to keep the soil in place, while an area with
sparse vegetation may increase the chances of landslides. The timber density of the Jecheon-si



region was divided into three classes: dense, moderate, and loose, as depicted in Fig. 9b. On
the other hand, it was reported that the likelihood of a landslide occurring is higher in newly
grown trees and less in older ones since aged trees have more roots (Lee and Min, 2001; Oh et
al., 2018). Thus, forest age was classified into seven classes: 61-70 years, 51-60 years, 41-50
years, 31-40 years, 21-30 years, 11-20 years, and <10 years, as depicted in Fig. 9c.

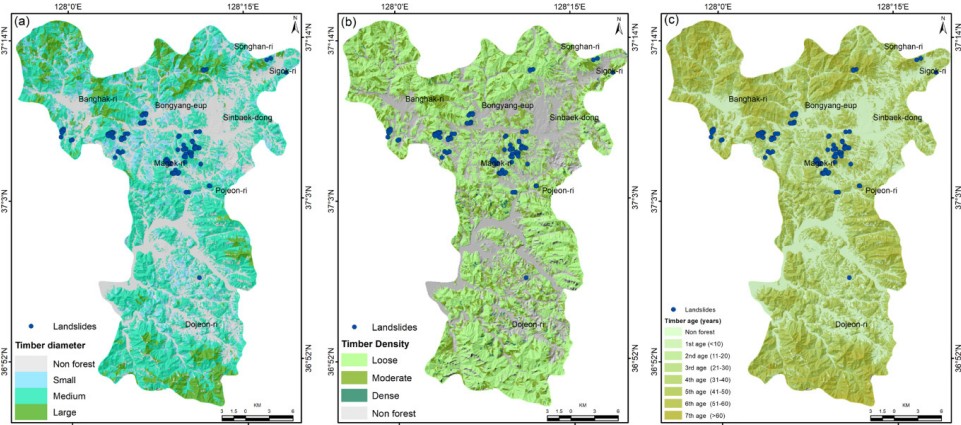

**Fig. 9** Forest predisposing factors of Jecheon-si region: **a** timber diameter, **b** timber density,
and **c** timber age.

**3.2 Methodology**
Landslide susceptibility was modeled using four statistical models: FR, IV, CF, and LR. Four
major steps were followed to achieve this goal: (a) a landslide inventory database was
generated to formulate and verify the required maps, (b) a GIS-based raster and vector database
of 18 predisposing factors were prepared to calculate the FR, IV, and CF values and to perform
subsequent analysis, (c) LR analysis was performed based on the dependent (landslide
inventory data) and independent variables (18 predisposing factors),  and (d) calculated LSIs
were validated using AUC and other statistical methods. This study aims to compare the most
widely used landslide susceptibility approaches and gain insight into their precision in
prediction capacities in susceptible zones.
**3.2.1 Frequency Ratio (FR) model**
The FR technique was proposed by Lee and Talib (2005) to explain the connection
between landslide locations and predisposing factors. The FR model estimates the probability
of an event or phenomenon occurring in a particular area (Lee and Pradhan, 2007). Since the





FR value represents the chance of occurrence, a higher value suggests a higher likelihood of a
landslide occurring and a greater associated risk. This method refers to the likelihood of an
incident based on data from previous landslides (Yilmaz, 2009). Subsequently, numerous
researchers (Agrawal and Dixit, 2022; Sonker et al., 2022; Dash et al., 2022; Huang et al.,
2020; Park et al., 2013; Choi et al., 2012; Yilmaz, 2009) round the world frequently used the
FR model in landslide susceptibility modeling.
The landslide density in each subclass/attributes layer of the predisposing factor was
calculated using Eq. (1) (Dash et al., 2022; Yilmaz, 2009):
$$FR_{ij} = \frac{NL_{ij}/NL_i}{N_{ij}/N_i} \qquad (1)$$
where $FR_{ij}$ represents the FR value of the $j^{th}$ attribute class in the $i^{th}$ predisposing factor, $NL_{ij}$
represents landslides in the $j^{th}$ attribute class, $NL_i$ represents landslides in the $i^{th}$ predisposing
factor (i.e., total landslides), $N_{ij}$ denotes the cells in $j^{th}$ attribute class, and $N_i$ denotes the cells
in the $i^{th}$ predisposing factor (i.e., total cells). The conditional probability principle supports the
FR approach. There were two possibilities: FR> 1 and FR <1 exhibiting high and weak
correlation, respectively (Huang et al., 2020). After that, the probability density ($FRP_{ij}$) of the
$j^{th}$ class of $i^{th}$ predisposing factor was calculated using Eq. (2) (Sonker et al., 2022):
$$FRP_{ij} = \frac{FR_{ij}}{\sum_{i=1}^{n} FR_{ij}} \qquad (2)$$
Finally, LSI was determined by summing all the $FRP_{ij}$ values based on the following equation:
$$LSI_{FR} = \sum_{i=1}^{n} FRP_{ij} \qquad (3)$$
where $LSI_{FR}$ represents the predicted LSI for each cell. The greater $LSI_{FR}$ value represents the
higher chances of landslide occurrence (Dash et al., 2022).
**3.2.2 Information Value (IV) model**
The IV model calculates the likelihood of a landslide in a specific location (Achu et al.,
2022; Zêzere et al., 2017). The model assigns a numerical value to each factor to determine the
overall risk of a landslide occurring. The model can identify areas that are most at risk of
landslides, allowing for implementing targeted mitigation measures to reduce the risk of
landslides. In general, the effect of various predisposing factors on the stability of LSI is
represented by information value (Huang et al., 2020). Yin and Yan (1988) first proposed this
model for landslide susceptibility analysis. Subsequently, various researchers (Dash et al.,
2022; Wang et al., 2019; Chen et al., 2014; Sarkar et al., 2013) used the IV model to map





landslide potential zones. The calculated information value determines the landslide
occurrence of each attribute class. The percentage of the total number of landslides for each
element was considered for computing IV values.
$$IV_{ij} = \ln\left(\frac{Denclass}{Denmap}\right) = ln\left(\frac{NL_{ij}/N_{ij}}{NL_i/N_i}\right) \qquad (4)$$
where $IV_{ij}$ represents the information value of the $j^{th}$ attribute class in the $i^{th}$ predisposing factor,
Denclass represents the landslide density of the $j^{th}$ attribute class, and Densmap represents the
landslide density of the $i^{th}$ predisposing factor. A positive IV value shows a high correlation
between the variable and the landslides. The zero IV shows no relationship between the
landslides and class attributes of predisposing factors, while a negative IV suggests a negative
relationship, i.e., variables favor slope stability (Chen et al., 2014). The LSI was determined
by adding the IV values for each predisposing attribute class as described below,
$$LSI_{IV} = \sum_{i=1}^{n} ln\left(\frac{NL_{ij}/N_{ij}}{NL_i/N_i}\right) \qquad (5)$$
The IV for each attribute class of influencing factor was determined based on the existence of
landslides in a certain class of influencing factor. Landslides are more likely to occur when IV
is high.

**3.2.3 Certainty Factor (CF) model**

The CF model describes the likelihood of a landslide occurring as related to the amount

of energy available to drive the landslide. The model considers various factors that influence
the energy available, such as lithological, soil, hydrological, topographical, and forest
characteristics. The model then uses these inputs to determine the likelihood of a landslide
occurring in a given area (Dash et al., 2022; Wang et al., 2019; Chen et al., 2019, Devkota et
al., 2013). The CF model is commonly used to assess unstable areas that have not experienced
landslides and the potential for further landslides in regions that have already experienced it
(Wang et al., 2019). The CF value was determined using Eq. (6) (Dash et al., 2022; Devkota et
al., 2013):
$$CF_{ij} = \begin{cases} \frac{pp_{ij}-pp_i}{pp_{ij}(1-pp_i)} & pp_{ij} \geq pp_i \\ \frac{pp_{ij}-pp_i}{pp_i(1-pp_{ij})} & pp_{ij} < pp_i \end{cases} \qquad (6)$$
where $pp_{ij}$ is the conditional probability in $j^{th}$ attribute class of $i^{th}$ predisposing factor and is
expressed as follows:



$pp_{ij} = \frac{NL_{ij}}{N_{ij}}$          (7)
$pp_i = \frac{NL_i}{N_i}$          (8)
The CF value ranges from -1 to 1. A higher chance of landslides is indicated by a positive CF
value, while a low likelihood is characterized by a negative CF value (Wang et al., 2019). The
CF value near zero does not indicate the certainty of landslide activities (Dash et al., 2022).
The LSI was calculated based on adding CF values of all predisposing factors using Eq. (9).
$LSI_{CF} = \sum CF_{ij}$          (9)
Generally, the high value of $LSI_{CF}$ represents greater chances of landslide susceptibility.

### 3.2.4 Multivariate Logistic Regression (LR) model

The LR model performs multivariate correlation analysis to investigate the connection
between independent and dependent variables. It is used to determine the probability of an
outcome based on various predictor variables (Devkota et al., 2013). A multivariate logistic
regression determines the proportional contribution of each independent variable to a
dependent variable's probability (Choi et al., 2012). It is used to determine the impact of several
independent variables on the likelihood of an event occurring. LR model is the most common
and reliable approach frequently used to evaluate the relationship between landslide inventory
and predisposing factors (Zhou et al., 2021; Aditian et al., 2018; Park et al., 2013; Yilmaz,
2009). Lee (2005) outlines the connection between the predisposition factors and the
occurrence of the phenomenon as,
$P = \frac{1}{1+e^{-Z}}$          (10)
where z is a linear combination of predisposing factors (i.e., independent variables) and p
denotes the likelihood of a landslide occurring. In an S-shaped curve, the probability varies
from 0 to 1. Equation (11) represents the linear combination of predisposing factors.
$Z = \beta_0 + \beta_1 x_1 + \beta_2 x_2 + \beta_3 x_3 + \cdots + \beta_n x_n$          (11)
where $\beta_0$ and $\beta_i$ represent the linear model's constant and regression coefficients, while $x_i$
represents the individual predisposing factor. The coefficient of each landslide predisposing
factor was determined using MATLAB. After that, the likelihood of landslides occurring for
each pixel was estimated using ArcGIS software based on the coefficient values.

### 3.2.5 Model performance

Model validation refers to evaluating a model's accuracy and reliability, typically by
comparing the model's predictions with in-situ observations. It can help identify any errors or





biases in the model and determine the extent to which it can accurately predict the outcome of
interest. In this paper, landslide density (Li), precision (P), MAE, MSE, RMSE, R-index, and
AUC were utilized to assess the effectiveness of the LSI models (He et al., 2021; Mandal et
al., 2021; Chen et al., 2019). The model validation was performed by examining the
relationship between the inventory and landslide susceptibility zones based on the R-index
analysis. The R-index was determined by applying Eq. (12) (Trinh et al., 2022; Shahabi and
Hashim, 2015):
$R = ({n_i}/{N_i}) / \Sigma ({n_i}/{N_i}) \times 100$       (12)
where $n_i$ represents the landslide inventory in the LSI zone, while $N_i$ represents the pixels in
the same LSI zone. On the other hand, the precision (P) parameter is widely used to validate
the predicted LSI. Thus, the precision of the predicted LSI was determined using Eq. (13)
(Ayalew et al., 2005).
$p = L_{hs}/L_T$       (13)
$L_{hs}$ represents the landslides in the severe to high LSI zone, while $L_T$ represents all landslides
in the region.

The MAE, MSE, and RMSE were also used to analyze the effectiveness of the LSI

model (Mandal et al., 2021; He et al., 2021; Chen et al., 2019). The RMSE and MSE measure
the forecasting errors of the model, whereas the MAE measures its generalization error
(Mandal et al., 2021). The RMSE, MAE, and MSE were determined using the following Eqs.

(14-16):

$RMSE = \sqrt{\frac{1}{n}\sum_{i=1}^{n}(l_{obs} - l_{pre})^2}$       (14)
$MAE = \frac{1}{n}\sum_{i=n}^{n}|(l_{obs} - l_{pre})|$       (15)
$MSE = \frac{1}{n}\sum_{i=n}^{n}(l_{obs} - l_{pre})^2$       (16)
where $l_{obs}$ denote the observed landslides, $l_{pre}$ is the calculated LSI values, and n represents the
inventory dataset (total samples) (He et al., 2021). A lower RMSE value denotes better model
performance.

Moreover, another common and widely adopted method frequently utilized to assess

the model performance in landslide susceptibility analysis is the Area Under the Receiver
Operating Characteristic Curve (AUC) (He et al., 2021, Pham et al., 2020). AUC measures
how accurately a predictive model can classify data points. Generally, the AUC curves are used
to calculate the precision of absence or presence prediction models (Shirzadi et al., 2017).



Higher AUC values indicate better performance, with a range of 0.5 to 1 (Chen et al., 2019;
Shahabi and Hashim, 2015).

**4. Results and Discussion**
**4.1 The spatial relationship between the predisposing factors and landslide inventory**

The landslide susceptibility of the Jecheon-si region was investigated using the FR, IV,
CF, and LR methods based on the landslide inventory data and 18 landslide influencing factors.
The relationship between the influencing factors viz. topographic slope, aspect, landforms
class, average shear-wave velocity, TPI, CI, TWI, TRI, plan curvature, profile curvature, SPI,
SL, surface lithology, soil thickness, timber density, timber age, soil type, timber diameter and
the landslide inventory locations were performed. After that, a spatial database was constructed
with a 5×5 m grid size. A relationship between the predisposing factors and landslide inventory
is depicted in Fig. 10 and Table 2.

**Table 2** Spatial relationships between landslide inventory location and the predisposing
factors.

| Predisposing Factors | Attributes | Area (km$^2$) | % of A | landslides | % of L | FRP$_{ij}$ | IV$_{ij}$ | CF$_{ij}$ |
|---|---|---|---|---|---|---|---|---|
| | Streams | 30.85 | 3.49 | 0 | 0.00 | 0.00 | 0.000 | -1.000 |
| | Midslope Drainages | 78.29 | 8.85 | 12 | 10.71 | 0.12 | 0.083 | 0.174 |
| | Upland Drainages | 10.06 | 1.14 | 4 | 3.57 | 0.32 | 0.497 | 0.681 |
| | Valleys | 79.67 | 9.01 | 1 | 0.89 | 0.01 | -1.004 | -0.901 |
| Landforms | Plains | 102.69 | 11.61 | 0 | 0.00 | 0.00 | 0.000 | -1.000 |
| | Open Slopes | 377.34 | 42.67 | 58 | 51.79 | 0.12 | 0.084 | 0.176 |
| | Upper Slopes | 70.01 | 7.92 | 12 | 10.71 | 0.14 | 0.131 | 0.261 |
| | Local Ridges | 2.80 | 0.32 | 0 | 0.00 | 0.00 | 0.000 | -1.000 |
| | Midslope Ridges | 76.62 | 8.66 | 16 | 14.29 | 0.17 | 0.217 | 0.394 |
| | High Ridges | 56.00 | 6.33 | 9 | 8.04 | 0.13 | 0.103 | 0.212 |
| | <180 | 45.58 | 5.16 | 0 | 0.00 | 0.00 | 0.000 | -1.000 |
| | 180-240 | 0.06 | 0.01 | 0 | 0.00 | 0.00 | 0.000 | -1.000 |
| | 240-300 | 0.03 | 0.00 | 0 | 0.00 | 0.00 | 0.000 | -1.000 |
| V$_s$$^{30}$ (m/sec) | 300-360 | 6.46 | 0.73 | 0 | 0.00 | 0.00 | 0.000 | -1.000 |
| | 360-490 | 27.78 | 3.14 | 0 | 0.00 | 0.00 | 0.000 | -1.000 |
| | 490-620 | 89.10 | 10.08 | 0 | 0.00 | 0.00 | 0.000 | -1.000 |
| | 620-760 | 254.68 | 28.82 | 35 | 31.25 | 0.45 | 0.035 | 0.078 |
| | >760 | 459.87 | 52.05 | 77 | 68.75 | 0.55 | 0.121 | 0.243 |
| | Metamorphic rocks | 295.70 | 33.44 | 36 | 32.14 | 0.39 | -0.017 | -0.040 |
| | Limestone | 147.01 | 16.62 | 1 | 0.89 | 0.02 | -1.270 | -0.946 |
| Rock Types | Granitic rocks | 407.68 | 46.10 | 75 | 66.96 | 0.59 | 0.162 | 0.312 |
| | Alluvium | 27.36 | 3.09 | 0 | 0.00 | 0.00 | 0.000 | -1.000 |
| | Sedimentary rocks | 6.58 | 0.74 | 0 | 0.00 | 0.00 | 0.000 | -1.000 |
| TRI | 0 - 0.43 | 158.71 | 17.95 | 0 | 0.00 | 0.00 | 0.000 | -1.000 |
| | 0.44 - 0.96 | 148.21 | 16.76 | 6 | 5.36 | 0.03 | -0.495 | -0.680 |





| | | | | | | | | |
|---|---|---|---|---|---|---|---|---|
| | 0.97 - 1.4 | 184.18 | 20.83 | 21 | 18.75 | 0.07 | -0.046 | -0.100 |
| | 1.5 - 1.9 | 169.91 | 19.21 | 27 | 24.11 | 0.10 | 0.099 | 0.203 |
| | 2 - 2.2 | 112.46 | 12.72 | 35 | 31.25 | 0.20 | 0.390 | 0.593 |
| | 2.3 - 2.6 | 64.51 | 7.29 | 12 | 10.71 | 0.12 | 0.167 | 0.319 |
| | 2.7 - 3 | 30.65 | 3.47 | 8 | 7.14 | 0.17 | 0.314 | 0.515 |
| | 3.1 - 3.6 | 11.97 | 1.35 | 2 | 1.79 | 0.11 | 0.120 | 0.242 |
| | 3.7 - 4.6 | 3.30 | 0.37 | 1 | 0.89 | 0.20 | 0.379 | 0.582 |
| | 4.7 - 14 | 0.43 | 0.05 | 0 | 0.00 | 0.00 | 0.000 | -1.000 |
| Plan Curvature | -51 - -10 | 0.05 | 0.01 | 0 | 0.00 | 0.00 | 0.000 | -1.000 |
| | -9 - -5 | 1.88 | 0.21 | 1 | 0.89 | 0.26 | 0.624 | 0.762 |
| | -4.9 - -2.5 | 17.86 | 2.02 | 6 | 5.36 | 0.16 | 0.424 | 0.623 |
| | -2.4 - -2 | 13.45 | 1.52 | 8 | 7.14 | 0.29 | 0.672 | 0.787 |
| | -1.9 - -0.5 | 143.57 | 16.23 | 32 | 28.57 | 0.11 | 0.245 | 0.432 |
| | -0.49 - 0.5 | 509.79 | 57.65 | 36 | 32.14 | 0.03 | -0.254 | -0.442 |
| | 0.51 - 2 | 156.94 | 17.75 | 22 | 19.64 | 0.07 | 0.044 | 0.097 |
| | 2.1 - 5 | 38.77 | 4.38 | 7 | 6.25 | 0.09 | 0.154 | 0.299 |
| | 5.1 - 10 | 2.01 | 0.23 | 0 | 0.00 | 0.00 | 0.000 | -1.000 |
| | 11 - 26 | 0.02 | 0.00 | 0 | 0.00 | 0.00 | 0.000 | -1.000 |
| Profile Curvature | -47 - -10 | 0.19 | 0.02 | 0 | 0.00 | 0.00 | 0.000 | -1.000 |
| | -9 - -5 | 2.93 | 0.33 | 0 | 0.00 | 0.00 | 0.000 | -1.000 |
| | -4.9 - -2 | 35.00 | 3.96 | 4 | 3.57 | 0.16 | -0.045 | -0.108 |
| | -1.9 - -0.5 | 131.70 | 14.89 | 25 | 22.32 | 0.27 | 0.176 | 0.333 |
| | -0.49 - 0.5 | 507.89 | 57.43 | 49 | 43.75 | 0.13 | -0.118 | -0.238 |
| | 0.51 - 2 | 169.50 | 19.17 | 29 | 25.89 | 0.24 | 0.131 | 0.260 |
| | 2.1 - 5 | 34.95 | 3.95 | 5 | 4.46 | 0.20 | 0.053 | 0.115 |
| | 5.1 - 10 | 1.99 | 0.23 | 0 | 0.00 | 0.00 | 0.000 | -1.000 |
| | 10.1 - 60 | 0.18 | 0.02 | 0 | 0.00 | 0.00 | 0.000 | -1.000 |
| Slope (degree) | <5 | 124.19 | 14.04 | 0 | 0.00 | 0.00 | 0.000 | -1.000 |
| | 5-10 | 71.44 | 8.08 | 2 | 1.79 | 0.04 | -0.656 | -0.779 |
| | 10-15 | 81.02 | 9.16 | 1 | 0.89 | 0.02 | -1.011 | -0.903 |
| | 15-20 | 104.14 | 11.78 | 6 | 5.36 | 0.07 | -0.342 | -0.545 |
| | 20-30 | 270.73 | 30.61 | 44 | 39.29 | 0.21 | 0.108 | 0.221 |
| | 30-40 | 192.86 | 21.81 | 49 | 43.75 | 0.33 | 0.302 | 0.502 |
| | 40-50 | 37.47 | 4.24 | 10 | 8.93 | 0.34 | 0.324 | 0.525 |
| | 50-60 | 2.39 | 0.27 | 0 | 0.00 | 0.00 | 0.000 | -1.000 |
| | 60-70 | 0.08 | 0.01 | 0 | 0.00 | 0.00 | 0.000 | -1.000 |
| | >70 | 0.01 | 0.00 | 0 | 0.00 | 0.00 | 0.000 | -1.000 |
| Aspect | Flat (-1) | 3.60 | 0.41 | 0 | 0.00 | 0.00 | 0.000 | -1.000 |
| | North (0-22.5) | 51.16 | 5.79 | 2 | 1.79 | 0.04 | -0.511 | -0.691 |
| | Northeast (22.5-67.5) | 107.14 | 12.12 | 3 | 2.68 | 0.03 | -0.655 | -0.779 |
| | East (67.5-112.5) | 106.91 | 12.09 | 22 | 19.64 | 0.20 | 0.211 | 0.385 |
| | Southeast (112.5-157.5) | 107.41 | 12.15 | 40 | 35.71 | 0.35 | 0.468 | 0.660 |
| | South (157.5-202.5) | 117.67 | 13.31 | 29 | 25.89 | 0.23 | 0.289 | 0.486 |
| | Southwest (202.5-247.5) | 115.99 | 13.12 | 8 | 7.14 | 0.07 | -0.264 | -0.455 |
| | West (247.5-292.5) | 114.44 | 12.94 | 5 | 4.46 | 0.04 | -0.462 | -0.655 |
| | Northwest (292.5-337.5) | 108.20 | 12.24 | 1 | 0.89 | 0.01 | -1.137 | -0.927 |
| | North (337.5-360) | 51.80 | 5.86 | 2 | 1.79 | 0.04 | -0.516 | -0.695 |
| SPI | 0 - 10 | 373.32 | 42.22 | 33 | 29.46 | 0.07 | -0.156 | -0.302 |
| | 10.1 - 25 | 199.70 | 22.58 | 29 | 25.89 | 0.12 | 0.059 | 0.128 |
| | 25.1 - 50 | 154.83 | 17.51 | 30 | 26.79 | 0.16 | 0.185 | 0.346 |
| | 50.1 - 75 | 64.28 | 7.27 | 12 | 10.71 | 0.15 | 0.168 | 0.322 |
| | 75.1 - 100 | 28.55 | 3.23 | 1 | 0.89 | 0.03 | -0.558 | -0.723 |
| | 101 - 200 | 29.73 | 3.36 | 4 | 3.57 | 0.11 | 0.026 | 0.059 |
| | 201 - 300 | 7.37 | 0.83 | 1 | 0.89 | 0.11 | 0.030 | 0.067 |
| | 301 - 500 | 6.66 | 0.75 | 2 | 1.79 | 0.25 | 0.375 | 0.578 |



| | | | | | | | |
|---|---|---|---|---|---|---|---|
| | 501 - 1000 | 6.88 | 0.78 | 0 | 0.00 | 0.00 | 0.000 | -1.000 |
| | >1000 | 13.01 | 1.47 | 0 | 0.00 | 0.00 | 0.000 | -1.000 |
| Slope Length (m) | 0 - 10 | 167.36 | 18.92 | 19 | 16.96 | 0.11 | -0.047 | -0.116 |
| | 10.1 - 20 | 83.19 | 9.41 | 13 | 11.61 | 0.15 | 0.091 | 0.190 |
| | 20.1 - 40 | 164.46 | 18.60 | 24 | 21.43 | 0.14 | 0.062 | 0.132 |
| | 40.1 - 60 | 121.18 | 13.70 | 22 | 19.64 | 0.17 | 0.156 | 0.302 |
| | 60.1 - 80 | 86.91 | 9.83 | 15 | 13.39 | 0.16 | 0.134 | 0.266 |
| | 80.1 - 100 | 63.00 | 7.12 | 7 | 6.25 | 0.11 | -0.057 | -0.140 |
| | 101 - 150 | 90.90 | 10.28 | 9 | 8.04 | 0.09 | -0.107 | -0.218 |
| | 151 - 200 | 43.87 | 4.96 | 1 | 0.89 | 0.02 | -0.745 | -0.820 |
| | 201 - 250 | 22.50 | 2.54 | 0 | 0.00 | 0.00 | 0.000 | -1.000 |
| | >250 | 40.96 | 4.63 | 2 | 1.79 | 0.05 | -0.414 | -0.615 |
| TPI | <-15 | 26.31 | 2.97 | 2 | 1.79 | 0.06 | -0.222 | -0.400 |
| | -14.9 - -10 | 59.49 | 6.73 | 11 | 9.82 | 0.14 | 0.164 | 0.315 |
| | -9.9 - -5 | 137.34 | 15.53 | 12 | 10.71 | 0.07 | -0.161 | -0.310 |
| | -4.99 - -2.5 | 111.57 | 12.62 | 11 | 9.82 | 0.07 | -0.109 | -0.222 |
| | -2.49 - 0 | 160.83 | 18.19 | 12 | 10.71 | 0.06 | -0.230 | -0.411 |
| | 0.001 - 2.5 | 111.70 | 12.63 | 12 | 10.71 | 0.08 | -0.071 | -0.179 |
| | 2.51 - 5 | 69.03 | 7.81 | 8 | 7.14 | 0.09 | -0.039 | -0.093 |
| | 5.01 - 10 | 99.88 | 11.29 | 24 | 21.43 | 0.18 | 0.278 | 0.473 |
| | 10.1 - 15 | 60.11 | 6.80 | 16 | 14.29 | 0.20 | 0.323 | 0.524 |
| | >15 | 48.07 | 5.44 | 4 | 3.57 | 0.06 | -0.182 | -0.343 |
| Convergence Index | <-15.0 | 17.75 | 2.01 | 1 | 0.89 | 0.05 | -0.352 | -0.555 |
| | -15.0 - -10 | 17.48 | 1.98 | 3 | 2.68 | 0.14 | 0.132 | 0.262 |
| | -10.0 - -5 | 52.22 | 5.90 | 16 | 14.29 | 0.25 | 0.384 | 0.587 |
| | -5.0 - -2 | 103.86 | 11.74 | 15 | 13.39 | 0.12 | 0.057 | 0.123 |
| | -2.0 - 0 | 247.22 | 27.96 | 29 | 25.89 | 0.09 | -0.033 | -0.080 |
| | 0 - 2 | 248.48 | 28.10 | 26 | 23.21 | 0.08 | -0.083 | -0.174 |
| | 2.0 - 5 | 111.27 | 12.58 | 15 | 13.39 | 0.11 | 0.027 | 0.061 |
| | 5.0 - 10 | 53.30 | 6.03 | 5 | 4.46 | 0.08 | -0.130 | -0.259 |
| | 10.0 - 15 | 16.56 | 1.87 | 2 | 1.79 | 0.10 | -0.021 | -0.049 |
| | >15 | 16.18 | 1.83 | 0 | 0.00 | 0.05 | 0.000 | -1.000 |
| | <-15.0 | 17.75 | 2.01 | 1 | 0.89 | 0.14 | -0.352 | -0.555 |
| Soil Type | Sandy loam | 132.07 | 14.95 | 36 | 32.14 | 0.21 | 0.333 | 0.535 |
| | Loam | 307.98 | 34.87 | 46 | 41.07 | 0.11 | 0.072 | 0.179 |
| | Silt loam | 66.65 | 7.55 | 2 | 1.79 | 0.02 | -0.625 | -0.763 |
| | Silty clay loam | 16.57 | 1.88 | 2 | 1.79 | 0.09 | -0.021 | -0.047 |
| | Silty clay | 2.17 | 0.25 | 0 | 0.00 | 0.00 | 0.000 | -1.000 |
| | Sandy clay loam | 1.18 | 0.13 | 0 | 0.00 | 0.00 | 0.000 | -1.000 |
| | Clay loam | 26.55 | 3.01 | 19 | 16.96 | 0.55 | 0.752 | 0.823 |
| | Silty | 0.07 | 0.01 | 0 | 0.00 | 0.00 | 0.000 | -1.000 |
| | No data | 330.00 | 37.36 | 7 | 6.25 | 0.02 | -0.777 | -0.833 |
| Soil Thickness (cm) | <20 | 89.07 | 10.07 | 0 | 0.00 | 0.00 | 0.000 | -1.000 |
| | 20-50 | 203.19 | 22.98 | 10 | 8.93 | 0.13 | -0.411 | -0.611 |
| | 50-100 | 461.05 | 52.14 | 92 | 82.14 | 0.53 | 0.197 | 0.365 |
| | >100 | 108.38 | 12.26 | 9 | 8.04 | 0.22 | -0.183 | -0.344 |
| | No Data | 22.64 | 2.56 | 1 | 0.89 | 0.12 | -0.457 | -0.651 |
| Timber Density | Loose | 468.53 | 52.98 | 86 | 76.79 | 0.53 | 0.161 | 0.310 |
| | Moderate | 143.98 | 16.28 | 21 | 18.75 | 0.42 | 0.061 | 0.132 |
| | Dense | 7.24 | 0.82 | 0 | 0.00 | 0.00 | 0.000 | -1.000 |
| | Non forest | 264.59 | 29.92 | 5 | 4.46 | 0.05 | -0.826 | -0.851 |
| Timber diameter | Small | 418.24 | 47.36 | 86 | 76.79 | 0.47 | 0.210 | 0.383 |
| | Medium | 96.33 | 10.91 | 18 | 16.07 | 0.42 | 0.168 | 0.321 |
| | Large | 105.18 | 11.91 | 3 | 2.68 | 0.06 | -0.648 | -0.775 |
| | Non forest | 263.38 | 29.82 | 5 | 4.46 | 0.04 | -0.825 | -0.850 |
| Timber age | 1st age | 0.03 | 0.00 | 0 | 0.00 | 0.00 | 0.000 | -1.000 |
| | 2nd age | 14.54 | 1.65 | 47 | 41.96 | 0.80 | 1.406 | 0.961 |
| | 3rd age | 91.11 | 10.32 | 3 | 2.68 | 0.01 | -0.586 | -0.740 |
| | 4th age | 168.55 | 19.09 | 6 | 5.36 | 0.01 | -0.552 | -0.719 |
| | 5th age | 249.71 | 28.28 | 39 | 34.82 | 0.04 | 0.090 | 0.188 |
| | 6th age | 84.44 | 9.56 | 12 | 10.71 | 0.04 | 0.049 | 0.108 |



| | | | | | | | | |
|---|---|---|---|---|---|---|---|---|
| | 7th age | 11.39 | 1.29 | 5 | 4.46 | 0.11 | 0.539 | 0.711 |
| | Non forest | 263.34 | 29.82 | 0 | 0.00 | 0.00 | 0.000 | -1.000 |
| TWI | 1.96 - 2.0 | 0.000156 | 0.000017 | 0 | 0 | 0.00 | 0.000 | -1.000 |
| | 2.01 - 4.0 | 20.82 | 2.35 | 4 | 3.57 | 0.39 | 0.181 | 0.340 |
| | 4.01 - 6.0 | 523.38 | 59.23 | 88 | 78.57 | 0.34 | 0.123 | 0.246 |
| | 6.01 - 8.0 | 187.93 | 21.27 | 18 | 16.07 | 0.20 | -0.122 | -0.244 |
| | 8.01 - 10.0 | 58.08 | 6.57 | 2 | 1.78 | 0.07 | -0.566 | -0.728 |
| | 10.1 - 12.0 | 26.97 | 3.05 | 0 | 0 | 0.00 | 0.000 | -1.000 |
| | 12.1 - 14.0 | 24.88 | 2.81 | 0 | 0 | 0.00 | 0.000 | -1.000 |
| | 14.1 - 16.0 | 19.39 | 2.19 | 0 | 0 | 0.00 | 0.000 | -1.000 |
| | 16.1 - 18.0 | 12.72 | 1.44 | 0 | 0 | 0.00 | 0.000 | -1.000 |
| | 18.1 - 23.3 | 9.33 | 1.05 | 0 | 0 | 0.00 | 0.000 | -1.000 |


The FR value indicates how closely landslides and a particular factor's attribute are
related, i.e., the higher the ratio, the stronger the association. The larger ratio indicated a more
significant association between the attribute of the given factor and occurrences of landslides
(Lee and Talib, 2005). The FR analysis exhibited that 40°- 50° slope angles have maximum
landslide occurrences (FR=2.11). In general, the slope shear stress changes as the slope angle
increases, which might result in landslides in a specific range (Chen et al., 2021; Park et al.,
2013). The FR analysis confirms the above statements. The other slope ranges, i.e., 5-10°
(FR=0.04), 10-15° (FR=0.02), 15-20° (FR=0.07), 20-30° (FR=0.21), and 30-40° (FR=0.33)
have decreasing order of the FR as depicted in Fig. 10. The maximum FR value (2.94) in the
slope aspect factor was observed in southeast-facing slopes. A landslide is likely also present
on the slopes facing east and south (FR = 1.62 and 1.95, respectively). For the other six aspects
classes, the FR value varies from 0.07 to 0.54, while zero FR is observed in the flat region. For
the plan and profile curvature, the plane curvature class of -2 to -2.5 is associated with a higher
FR value (4.70) and is most susceptible to slope failure. On the other hand, the -0.5 to -1.9 class
of the profile curvature was associated with higher FR (1.50) and had the highest probability
of landslide occurrences. The convergence index presented the higher FR value associated with
a subclass ranging from -5 to -10 (FR=2.42). The FR analysis of the landform class showed
that the upland drainages were more susceptible to slope failure (FR= 3.14), and local ridges
were less vulnerable (FR=0). On the other hand, a high FR value was associated with the TPI
class of 10.1-15 (FR=2.10). Regarding the TRI factor, the higher FR value was related to the
TRI classes of 1.5 to 4.6, as depicted in Fig. 10.
Figure 10 depicts the correlation between the rate of landslide occurrence and the SPI.
It was observed that the SPI subclass of 300-500 was associated with higher FR (2.40) and was
most susceptible to slope failure. A higher SPI value often indicates a higher likelihood of a
landslide occurring. The TWI subclass of 2.0-4.0 was associated with higher FR (1.46),





representing higher landslide occurrences. For slope length, the higher FR value was observed
in the slope length of <100m, while relatively low FR values were associated with higher slope
length (Fig. 10). The surface geology showed that granitic and metamorphic rocks with FR
values of 1.45 and 0.96 were the rock outcrop most likely to experience landslides. On the other
hand, the average shear wave velocity indicates that site class C1 ($V_s^{30}\sim$ 620-760 m/sec) and B
($V_s^{30}$ >760 m/sec) are vulnerable to slope failure with FR values of 1.08 and 1.32, respectively.

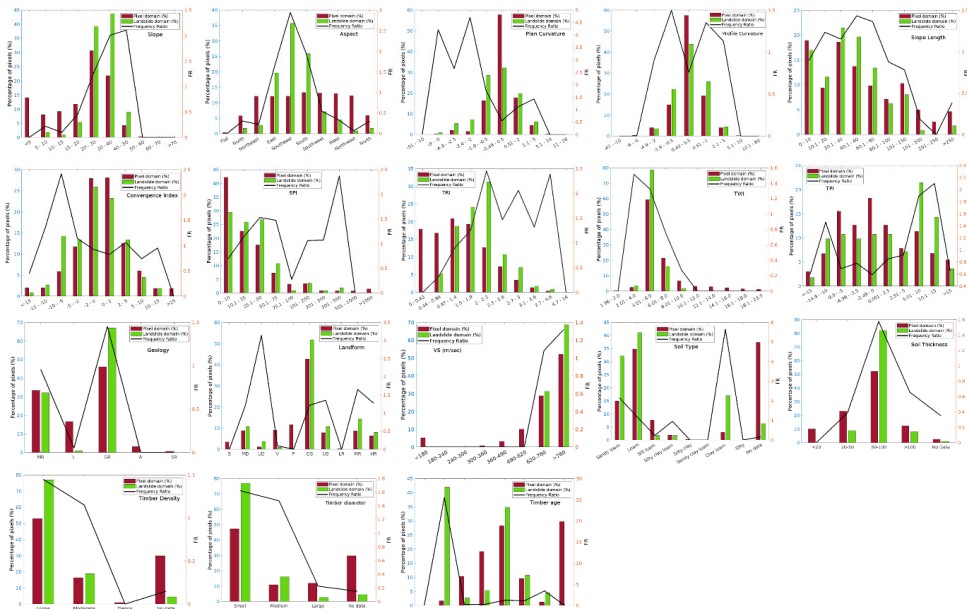


**Fig. 10** The spatial relationship between the landslide locations and the predisposing factors.

Regarding the soil, the spatial relationship between the landslide locations and the eight
soil types shows a clear difference in relevance to the likelihood of landslides. It was observed
that the clay loam soil is most likely to slide (FR= 5.65), while silty clay and sandy clay are
relatively stable. On the other hand, soil depth also plays a role in the water content of the soil,
as soils with deeper depth can hold more water and become saturated more easily. The higher
FR values were observed in the soil depth of >50 cm. The study explored the relationships
between timber diameter, density, and age as factors associated with forest type and landslides.
The FR was high in low-density forest areas (FR= 1.62). In the case of the timber age class,
the FR is higher for 10 to 20-year-old forest areas (FR=0.80). Conversely, small-diameter
timber was associated with higher landslide probability (FR= 25.48).




## 4.2. Multi-collinearity analysis of predisposing factors

The statistical phenomenon of multi-collinearity arises when two or more predictor
variables in a regression model are strongly correlated (Zhou et al., 2021). The multi-
collinearity of predisposing factors will negatively affect the model's outcomes, reducing its
predictive power or perhaps making it fail. Therefore, understanding collinearity among the
predisposing factors, that is, whether there is a linear correlation among the independent
predisposing factors, is critical before performing the LR modeling. Here, we determined the
multi-collinearity among predisposition factors using the variance inflation factor (VIF) and
tolerance (Chen and Chen, 2021).
$$TOL = \frac{1}{VIF} \tag{17}$$
$$VIF = \frac{1}{1 - R_i^2} \tag{18}$$
where $R_i^2$ represents the coefficient of determination of regression of variable 'i' on all other
variables (Hong et al., 2020). Table 3 illustrates the results of the multi-collinearity analysis
for all variables that met the threshold values (VIF<10 or tolerance > 0.1) (Zhang et al., 2020;
Kadavi et al., 2019). According to the results of the multi-collinearity diagnostics tests (Table
3), TRI has the highest VIF (6.483) and lowest tolerances (0.154), which are far from the
critical values. Subsequently, these eighteen variables were selected for LSI modeling.

**Table 3** Regression coefficient and collinearity of the landslide-predisposing factors.

| Predisposing factors | β | Collinearity statistics | |
|---|---|---|---|
| | | Tolerance | VIF |
| Slope | 9.527 | 0.158 | 6.341 |
| Aspect | 10.547 | 0.890 | 1.123 |
| Convergence index | -1.095 | 0.687 | 1.455 |
| Plane curvature | 3.093 | 0.607 | 1.646 |
| Profile curvature | 4.089 | 0.870 | 1.149 |
| TRI | -4.953 | 0.154 | 6.483 |
| TPI | 2.679 | 0.780 | 1.282 |
| Landform | 13.603 | 0.529 | 1.890 |
| Slope Length | 16.704 | 0.865 | 1.157 |
| TWI | -5.004 | 0.547 | 1.827 |
| SPI | 3.956 | 0.810 | 1.235 |
| Geology | 6.223 | 0.897 | 1.115 |
| $V_s^{30}$ | 2.381 | 0.488 | 2.049 |
| Soil type | 6.696 | 0.701 | 1.427 |
| Soil thickness | 4.654 | 0.889 | 1.125 |





| Timber density | 2.897 | 0.212 | 4.722 |
| Timber diameter | -0.208 | 0.278 | 3.593 |
| Timber age | 2.244 | 0.777 | 1.287 |
| Intercept | -16.500 | - | - |


**4.3 Landslide susceptibility index (LSI) based on the FR, IV, CF and LR models**

The FR, IV, CF and LR models were independently constructed to determine the LSI
using Eqs. 3, 5, 9 and 11. Here, we used ArcGIS software to develop the LSIs based on four
statistical models. The calculated LSI values for the FR, IV, CF and LR models vary from 0.47
to 6.06, -10.15 to 5.06, -14.56 to 7.66, and 0-1, respectively. High LSI values represent more
susceptibility to landslides, whilst low LSI levels suggest less susceptibility to landslides (Dash
et al., 2022). The LSI value was then normalized using Eq. (19) to understand the effectiveness
of predicted LSI with the topographic and landslide characteristics.

$$LSI_{nm} = \frac{LSI_o - LSI_{\min}}{LSI_{\max} - LSI_{min}} \tag{19}$$

where $LSI_{nm}$ represents the normalized landslide susceptibility index for FR, IV, CF and LR
models, $LSI_o$ represents the original LSI value, and $LSI_{max}$ & $LSI_{min}$ represent the minimum
and maximum LSI value. The normalized LSI values for the FR, IV, CF, and LR models are
depicted in Fig. 11. The spatial distribution of LSI values derived based on the FR, IV, and CF
models (Figs. 11a-c) was somewhat comparable. On the other hand, the LSI distribution
computed through the LR model differs from the other models (Fig. 11d). The distribution of
LSI showed that high-elevation areas have a higher likelihood of experiencing landslides. The
region's northern, southern and central parts, with steep slopes surrounding the valley, were
identified as the most vulnerable region. On the other hand, relatively flat areas exhibited low
landslide potential. It was noted that the majority of the land cover in this region had a low
density of forests.

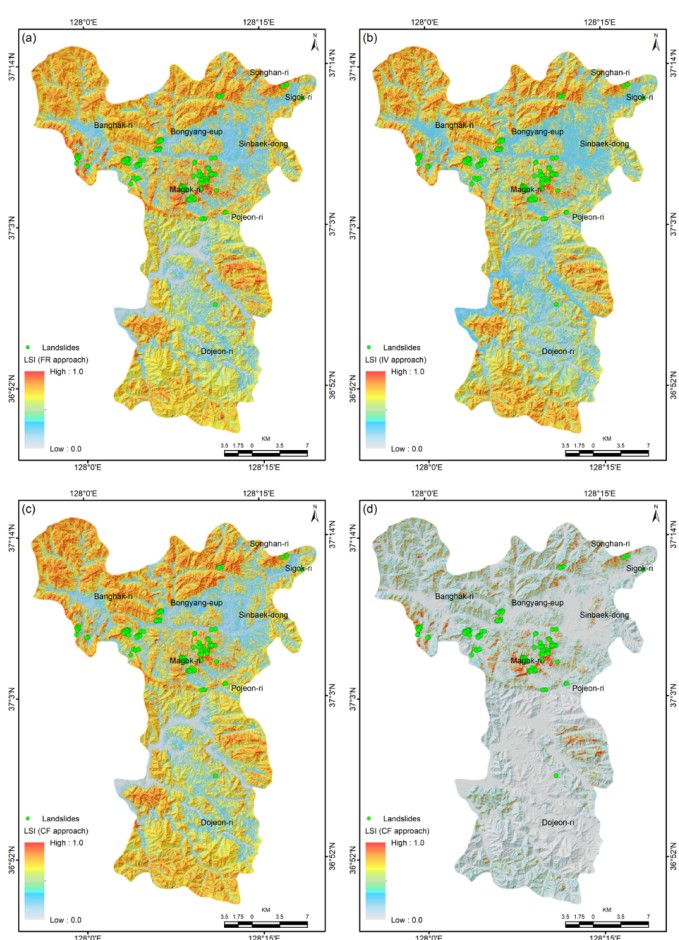

**Fig. 11** Landslide susceptibility index of the Jecheon-si region based on the **a** FR, **b** IV, **c** CF, and **d** LR model.

Table 4 illustrates the MSE, RMSE, MAE, and AUC values of the FR, IV, CF and LR models. The AUC values for the FR, IV, CF, and LR models were found to be 0.889, 0.872, 0.877, and 0.912, respectively. All four of the models' AUC values were greater than 0.80, showing that the LSI models had strong prediction abilities. Based on the inventory datasets, the models' accuracy was further examined using RMSE, MSE, and MAE. The outcome demonstrates that the FR, CF and IV models had the lowest RMSE, MSE, and MAE values. On the other hand, the LR model had higher MAE, MSE, and RMSE values, signifying lower prediction accuracy than other models. The LR model, however, performs better than other



models in terms of AUC value. Therefore, selecting an appropriate model for landslide
susceptibility mapping is difficult even though the performances and prediction accuracy of all
the discussed models were acceptable.

**Table 4** Validation of models by AUC, RMSE, MSE, and MAE.

| Models | AUC | MAE | MSE | RMSE |
|---|---|---|---|---|
| FR | 0.889 | 0.281 | 0.087 | 0.295 |
| IV | 0.872 | 0.238 | 0.059 | 0.243 |
| CF | 0.877 | 0.226 | 0.064 | 0.252 |
| LR | 0.912 | 0.272 | 0.147 | 0.385 |


In this study, we used a different method to evaluate the results of LSI. The approach

uses a high-resolution DEM, aerial photos, and drone images to determine whether a landslide
disaster is likely in the predicted very high susceptibility area (He et al., 2021). We also used
the 1D elevation profile to check whether the predicted LSI distribution is consistent with the
topographic and landslide characteristics. To better display the experiment and evaluate the
model accuracy, we selected recent landslide sites that have never previously experienced
landslides. The aerial photo was acquired from the NGII web portal (https://map.ngii.go.kr/)
for 2020 to 2021, and the drone survey was conducted in August 2020. Figure 12 depicts the
LSI distribution and the landslide area on a dronograph and elevation profile from the landslide
source area to the landslide deposition zone. The landslide-affected regions are clearly visible
in both the drone and the aerial photos. The predicted LSI value based on the FR, IV, and CF
models was found to be very high in both the crown and the landslide deposit zone (Fig. 12f).
In contrast, the LSI predicted by the LR model was low in the landslide deposit zone and
moderate in the crown zone (Fig. 12f). The four applied models were found to be able to predict
the location of the landslide precisely; however, they are not consistent with the landslide and
topography characteristics. To overcome this issue, we put forth a hybrid integrated strategy to
verify that the LSI derived using an integrated approach is consistent with topography and
landslide characteristics.




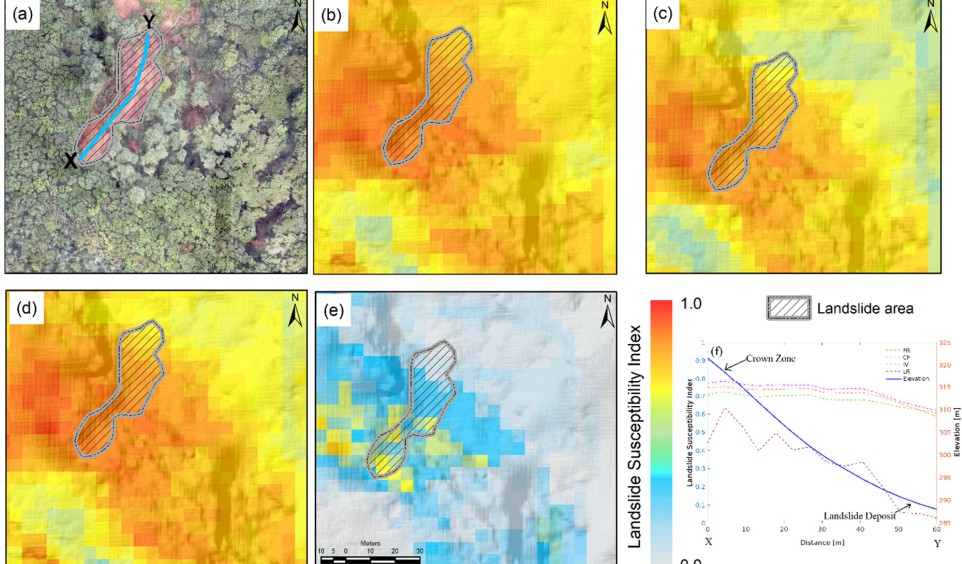

Fig. 12 Spatial characteristics of predicted LSIs: **a** Drone image acquired in August 2020, **b**
LSI based on the FR model, **c** LSI based on the IV model, **d** LSI based on the CF model,
**e** LSI based on the FR model and **f** elevation profile and LSI distribution from the landslide
source area to landslide deposit zone.

**4.4 LSI based on a hybrid integrated approach**

A combination of different models is one of the options for improving model accuracy.
Therefore, we integrated the above four models using Eq. (20).

$$LSI_{intergrated} = w_{0.25}.LSI_{FR} + w_{0.25}.LSI_{IV} + w_{0.25}.LSI_{CF} + w_{0.25}.LSI_{LR} \qquad (20)$$

where $LSI_{FR}$, $LSI_{IV}$, $LSI_{CF}$, and $LSI_{LR}$ represent the normalized LSI of each model, and w
represents the weight of each LSI model. The spatial distribution of LSI estimated using the
integrated technique is shown in Fig. 13. It was observed that the LSI predicted through the
integrated approach was consistent with the topographic profile and landslide characteristics,
which was not in the earlier case. For example, a high LSI value was observed in the landslide
source area, while a low LSI value was observed in the landslide deposit zone (Figs. 13b′-d').
Therefore, the LSI calculated through the hybrid integrated approach was used for further
analysis.





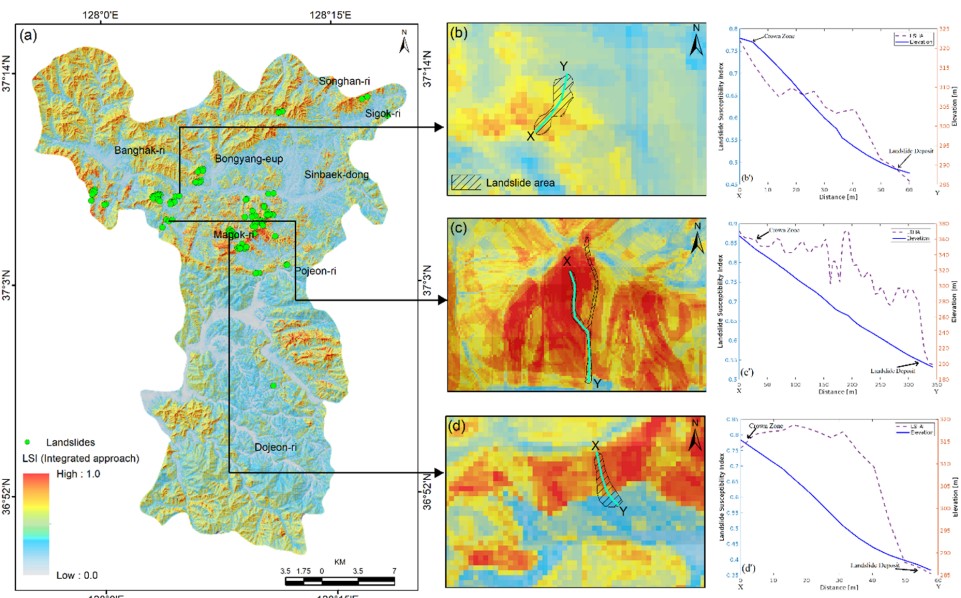

**Fig. 13** LSI based on the hybrid integrated approach: **a** spatial distribution of LSI in the Jecheon-si region, **b-d** the details of LSI distribution of three recent past landslide events, and **b'-d'** elevation profile and LSI distribution from the landslide source area to landslide deposit zone at different landslide sites.

The LSI was categorized using Jenks natural breaks (Huang and Zhao, 2018) into five microzones: unlikely, low, moderate, high, and severe. The landslide hazard microzonation map (Fig. 14a) shows that 2.73% of the total areas are classified as severe susceptibility (SS). The areas classified as high, moderate, low, and unlikely zones were 14.94%, 40.31%, 30.20%, and 11.82%, respectively. The severe susceptibility region was observed to contain 41.96% of landslides, while the area of the unlikely zone is associated with zero landslides. A potential landslide-prone area can be seen on the map in the central part of the study area, i.e., the Magok-ri region. The landslide hazard microzonation map derived through the hybrid integrated approach is accurate and relevant since more landslides occur in the zones with the highest susceptibility.



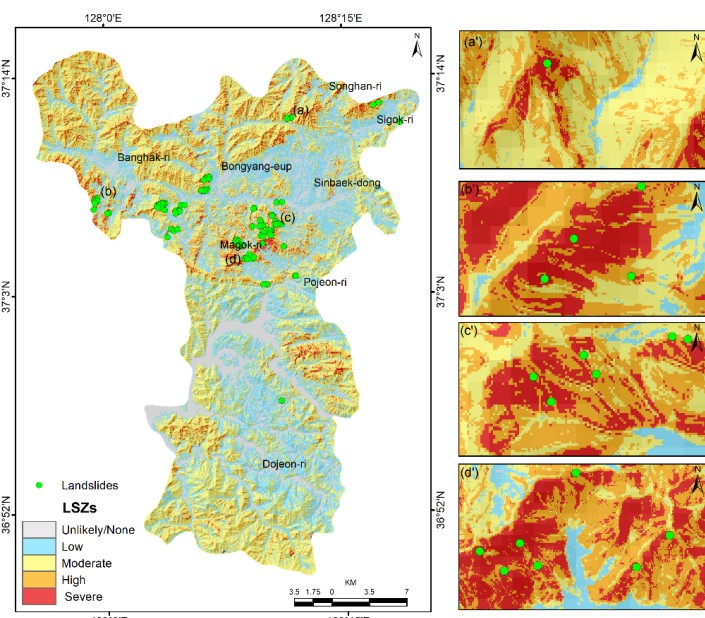

**Fig. 14 a** Landslide hazard microzonation map of the Jecheon-si region using a hybrid
Integrated approach, while subplots **a'-d'** represent the region's detailed LSI and
associated landslide inventory.

### 4.5 Validation of landslide hazard microzonation map based on landslide inventory and in-situ observations

Evaluating the accuracy assessment between susceptibility classes and actual landslide observations is essential, as past instability evidence often serves as the best guide for predicting future behavior in the locality. Thus, the integrated landslide hazard microzonation map was verified based on the reported and in-situ observations. We used the landslide inventory database to calculate the effectiveness of the integrated LSI method and determine the precision of the susceptibility index. Subsequently, the landslide density ($L_i$), precision (P), MAE, MSE, RMSE, R-index, and AUC have been calculated to validate the outcome. Table 5 illustrates detailed landslide frequencies for each susceptibility zone. It was observed that the $L_i$ values increased gradually from the unlikely to the severe susceptible classes. Additionally, $L_i$ values vary considerably between classes. Therefore, it can be said that the developed hazard microzonation map indicates reasonable hazard classes. We also assessed the validity of the calculated LSI based on the P-value, which is the difference between the slide area in the upper low (severe to high) and the total area of the slide. The precision of our proposed methods was



determined to be 88.3%, which is deemed acceptable for identifying the likelihood of landslide-
prone regions in this area. In addition, the R-index results indicate that the LSM has a very high
prediction accuracy.

The hybrid integrated LSI model was further examined using MSE, MAE, and RMSE

with the landslide inventory data. The MSE, MAE, and RMSE values for the integrated models
are 0.25, 0.08, & 0.28, respectively, exhibiting good consistency with the in-situ observations.
On the other hand, correct classification percentages (for 0.5 cut-off value) are also calculated
to assess the LSI's sensitivity (Gorum et al., 2008). It is exhibited that the integrated models
have a prediction capacity of 94.6% (Fig. 15a).

**Table 5** Accuracy statistics of the landslide hazard zonation map.

| LSZM | $P_i$ | % $P_i$ | $L_i$ | % $L_i$ | Landslide Density | R-Index | P | MAE | MSE | RMSE |
|---|---|---|---|---|---|---|---|---|---|---|
| Unlikely | 4174356 | 11.82 | 0 | 0.00 | 0.00 | 0.00 | | | | |
| Low | 10661338 | 30.20 | 2 | 1.79 | 0.06 | 0.31 | | | | |
| Moderate | 14230579 | 40.31 | 11 | 9.82 | 0.24 | 1.30 | 88.3 | 0.25 | 0.08 | 0.28 |
| High | 5272713 | 14.94 | 52 | 46.43 | 3.11 | 16.53 | | | | |
| Severe | 962462 | 2.73 | 47 | 41.96 | 15.39 | 81.86 | | | | |

*LSZM-landslide susceptibility zonation map

The LSI models were also evaluated through the ROC curve analysis. The result of the

ROC curve test is shown in Fig. 15b. An AUC value of 0.7 or more indicates good predictive
performance (Mandal et al., 2021). The AUC values obtained from the FR, IV, CF, LR and
integrated model are 0.889, 0.872, 0.877, 0.912, and 0.909, respectively, which suggests a high
landslide prediction rate. Therefore, the zones of high and severe landslide susceptibility
identified by all the models indicate the degree of field instability. However, the LSI predicted
based on the integrated method was consistent with the topographic and landslide
characteristics, suggesting more reliable and appropriate outcomes than other models.





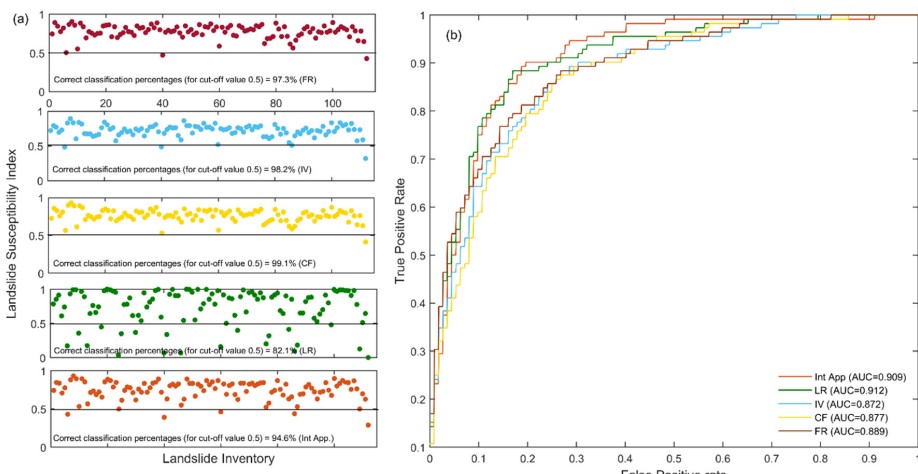


**Fig. 15 a** The estimated LSI of the landslide inventory datasets with correct classification percentages, and **b** Model performance based on the AUC.

**5. Conclusion**

Landslide susceptibility analysis requires a detailed database of landslide inventory and influencing factors. The high-resolution spatial database, comprising 18 predisposing factors, including topography, hydrology, lithology, soil, and forest aspects, was utilized to predict LSI for the Jecheon-si region of South Korea. It was observed that the IV, CF and FR models predicted higher LSI values in both the landslide source area and the landslide deposit zone. In contrast, the LR model predicted moderate LSI values in the landslide source area and lower LSI values in the landslide deposit zone. So, the four applied models can successfully and reliably predict the susceptible zones; however, the predicted LSI distributions were not always consistent with topographic and landslide characteristics. To overcome this issue, we proposed a hybrid integrated approach for better performance. It was observed that the LSI predicted through a hybrid integrated approach was consistent with the topographic and landslide characteristics, which were not present in the earlier models. After that, the LSI calculated through the hybrid integrated approach was classified into five landslide susceptibility microzones: unlikely, low, medium, high, and severe. It was observed that most landslides occurred in the very high to severe susceptible zones, with a gradual decrease toward lower susceptibility zones, which indicates the LSZ agreed with field instability. The precision results (i.e., AUC=0.906, MSE=0.0.25, MSE=0.08, RMSE=0.28, P=88.3%) suggest that the hybrid integrated approach would be useful for landcover planning and landslide induces disaster mitigation purposes. In addition, this research methodology will be helpful for the local



government's disaster preventative measures and be a useful, practical reference for predicting
the risk of landslides on the Korean Peninsula.

**Acknowledgment**
This research work was funded by the National Research Foundation, Korea (NRF) under
Grant [2021R1C1C2010999]. This research work was also funded by the National Research
Foundation of Korea (NRF) grant funded by the Korea government (MSIT) (NRF-
2021R1C1C2003316).

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
