# Peer review of "Landslide Hazard Microzonation Using a Hybrid Integrated Approach to Reduce 2 Disaster Risk: A Case Study of Jecheon, South Korea 3 Jae-Joon Lee1, Manik Das Adhikari2, Moon-Soo Song2, Sang-Guk Yum2\* 4 5 6 1 Department of Fire Safety Engineeri"

_EGUsphere, 2025_

## Author Comment (AC1)

**Manuscript ID: egusphere-2025-1169**

My co-authors and I would like to express our gratitude to the reviewers for their constructive feedback and suggestions for strengthening our research. The changes we have made to the attached file in response to such feedback and suggestions have been highlighted in blue to facilitate their identification. I would also like to offer my apologies for the length of time it took us to prepare this response.

**Response to Reviewer #1**

**Overall Observations:** This study utilizes 112 landslide inventory points in Jecheon, South Korea, to generate landslide susceptibility maps using four commonly applied models: Frequency Ratio (FR), Certainty Factor (CF), Logistic Regression (LR), and Information Value (IV). The spatial consistency of the predicted Landslide Susceptibility Index (LSI) with observed landslide profiles was assessed using high-resolution aerial and drone imagery. To address observed spatial inconsistencies, a hybrid integrated model was proposed, resulting in a microzonation hazard map that reportedly shows improved predictive performance, with high precision metrics such as an AUC of 0.906. However, the manuscript has several critical flaws.

Thank you for your insightful review. We are very grateful for your constructive suggestions, which have greatly helped improve the preprint.

**Comment 1:** First, the landslide distribution map is of low quality and the number of samples is insufficient to represent the actual pattern of landslide development in the region, especially in the context of rainfall-induced clustered events. The limited sample size also raises concerns about severe overfitting, which can lead to high accuracy on training data but poor generalization in real-world applications.

**Response:** Thank you for your insightful observation. In the present study, the inventory database was created using high-resolution aerial imagery (obtained from http://map.ngii.go.kr/ms/map/), historical Google Earth imagery, field investigation and compilation of available inventory data from the Korea Forest Service (KFS). As suggested, we further rigorously used historical Google Earth imagery and aerial photographs to update the landslide inventory data. Subsequently, we prepared an updated landslide inventory ($n$=160) map of the Jecheon-si region, as shown in Fig. 5. Additional, in the revised manuscript, to better understand the spatial distribution of landslide events in this region, we analyzed the long-term (2000-2019) maximum daily rainfall intensities using TRMM (Tropical Rainfall

Measuring Mission, https://gpm.nasa.gov/missions/trmm) datasets processed in Google Earth Engine. Results revealed that maximum rainfall intensity ranged from 43.74 to 56.7 mm/day across the region, with the central and northern parts exhibiting significantly higher values (Fig. 5). This pattern aligns with the concentration of landslide events in the northern and central parts of the study region, indicating higher rainfall intensity is a critical factor controlling the spatial distribution of landslides. Subsequently, long-term rainfall intensity was utilized as a triggering factor for the landslide hazard microzonation analysis.

[Figure]

Figure 5. Maximum rainfall intensity (mm/day) distribution of the region for the period of 2000 to 2019, and spatial distribution of landslide inventory.

Regarding potential overfitting, the spatial clustering and limited inventory size indeed limit the suitability of data-intensive ML models. For this reason, we adopted widely used

interpretable statistical approaches, FR, IV, CF, and LR, which are robust under data-scarce conditions (Ding et al., 2025; Xie et al., 2017). To train and validate the LSI models, we randomly split the landslide inventory dataset into training and testing, i.e., 70% and 30%, respectively (Nguyen et al., 2021; Zhou et al., 2021b; Aditian et al., 2018). Model performance was evaluated using AUC, MAE, MSE, and RMSE (Table 4). It was observed that all models achieved AUC > 0.85 for both training and testing, indicating high discriminative ability. The LR model yielded the highest AUC (0.901 training, 0.930 testing), while FR, IV, and CF achieved slightly lower but comparable results (0.858-0.910). Although LR displayed slightly higher MAE, MSE, and RMSE values, all models demonstrated robust predictive accuracy without clear signs of overfitting.

Table 4. Validation of models by AUC, RMSE, MSE, and MAE.

| Models | Traing | | | | Validation | | | |
|--------|--------|--------|--------|--------|--------|--------|--------|--------|
| | AUC | MAE | MSE | RMSE | AUC | MAE | MSE | RMSE |
| FR | 0.879 | 0.291 | 0.092 | 0.303 | 0.905 | 0.267 | 0.078 | 0.279 |
| IV | 0.858 | 0.248 | 0.068 | 0.261 | 0.910 | 0.220 | 0.053 | 0.230 |
| CF | 0.861 | 0.236 | 0.063 | 0.252 | 0.917 | 0.204 | 0.046 | 0.215 |
| LR | 0.901 | 0.306 | 0.174 | 0.417 | 0.930 | 0.238 | 0.125 | 0.354 |

Additionally, to explicitly assess model generalization in real-world scenarios, we tested spatial characteristics of the model outputs against recent landslide sites not included in the inventory by utilizing high-resolution LiDAR DEM, aerial photographs (NGII, 2020-2021), and drone surveys (August 2020). It was observed that even though the statistical model evaluation revealed high AUC values and acceptable statistical metrics, spatial analysis revealed inconsistencies of LSI distribution in landslide source and deposition zones (Figs. 14 and S1). For instance, FR, IV, and CF models predicted high LSI values in both the crown (~0.75) and deposit (~0.69) zones, whereas LR predicted moderate susceptibility in the crown (~0.62) but very low in the deposit (~0.05). To overcome this issue, we put forth a hybrid integrated strategy to verify whether the LSI derived from the integrated approach aligns with topography, geomorphic features, and landslide characteristics. It was observed that the LSI predicted through the integrated approach resolves spatial inconsistencies by combining the strengths of multiple models, which was not in the earlier case. For example, a high LSI value was observed in the landslide source area (i.e., 0.75 to 0.9), while a comparatively lower LSI value was observed in the landslide deposit zone (i.e., 0.35-0.55). This cross-temporal

validation confirms that the proposed models are not overfit to historical events and can reliably predict landslide-prone zones under varying topographic and geomorphic settings. Moreover, the hybrid integrated strategy leverages the complementary strengths of the individual models, yielding improved spatial coherence and practical applicability for real-world landslide hazard management and proactive mitigation planning.

[Figure]

Figure 14. Spatial characteristics of predicted LSIs: (a) Drone image captured in August 2020, (b) LSI based on the FR model, (c) LSI based on the IV model, (d) LSI based on the CF model, (e) LSI based on the LR model and (f) elevation profile and LSI distribution from the landslide source area to landslide deposit zone (additional experimental site is illustrated in Figure S1 (electronic supplementary data)).

[Figure]

Figure S1. LSI based on the hybrid integrated approach: (a) spatial distribution of LSI in the Jecheon-si region, (b-d) the details of LSI distribution of three recent past landslide events, and (b'-d') elevation profile and LSI distribution from the landslide source area to the deposition zone at different landslide sites.

In the revision, we incorporated the above discussion to address concerns about sample size, minimize the risk of overfitting, and enhance the generalization capability of the proposed hybrid integrated LSI model.

**Comment 2:** Additionally, the manuscript lacks meaningful and insightful scientific discussion. The spatial inconsistency between predicted susceptibility and actual landslide characteristics is acknowledged but not adequately analyzed or explained.

**Response:** Thank you for your comment and for acknowledging the discussion regarding the spatial inconsistency between predicted susceptibility and actual landslide characteristics. In the revision, we incorporated additional analyses and explanations to interpret these discrepancies better, as highlighted below,

Figure 13 illustrates the ROC-AUC values of the FR, IV, CF and LR models, which reveal that all four models demonstrate strong discriminatory power, with AUC values consistently exceeding 0.85, indicating high model reliability. It was observed that the LR model achieved the highest AUC values for both training (0.901) and testing (0.930) datasets,

signifying its superior generalization capability and robustness in capturing the non-linear relationships between landslide occurrence and contributing factors. The FR model yielded AUCs of 0.879 (training) and 0.905 (testing), followed closely by IV (0.858 and 0.910) and CF (0.861 and 0.917), reflecting their competency in bivariate and heuristic-based spatial correlation assessments. Nevertheless, all models yielded acceptable and comparable prediction accuracies in both the training and testing datasets, indicating their robustness for landslide susceptibility mapping. Therefore, selecting an appropriate model for landslide susceptibility mapping is difficult, even though the performances and prediction accuracy of all the discussed models were acceptable.

[Figure]

Figure 13. ROC curves and corresponding AUC values for four LSI models: (a) LR, (b) FR, (c) IV and (d) CF models [Note: Blue and red curves denote training and testing datasets, respectively, used for model evaluation. The block dotted line represents the random guess].

In this study, we used an alternative approach to evaluate the LSI results and explicitly assess model generalization in real-world scenarios. The approach integrates a high-resolution DEM (5×5m), aerial photos, and drone images of recent past landslide events (not included in the inventory datasets) to verify whether the unmapped landslide sites fall within the predicted

very high susceptibility zones (He et al., 2021). We also used the 1D elevation profile to check whether the predicted LSI distribution is consistent with the topographic, geomorphic, and landslide characteristics. Consequently, we selected recent event sites that have not previously experienced landslides to better demonstrate the experimental analysis and evaluate the model accuracy. For these purposes, we acquired aerial photos from the NGII web portal (https://map.ngii.go.kr/) for 2020 to 2021, and the drone survey was conducted in August 2020. Figure 14 depicts the predicted LSI distributions and the landslide area marked on a dronograph and elevation profile from the landslide source area to the landslide deposition zone. The landslide-affected regions are clearly visible in the drone imagery (Figs. 14a). The predicted LSI value based on the FR, IV, and CF models was found to be very high in both the crown (i.e., ~0.75) and the landslide deposit zone (i.e., ~0.69) (Fig. 14f). In contrast, the LSI predicted by the LR model was low in the landslide deposit zone (i.e., ~0.05) and moderate in the crown zone (i.e., ~0.62) (Fig. 14f). A similar experimental results was also found at different landslide event as depicted in Fig. S1 (supplementary data). It was observed that all models could identify the landslide source area precisely; however, differences in susceptibility distribution highlight that spatial consistency with topography varies by model.

[Figure]

Figure 14. Spatial characteristics of predicted LSIs: (a) Drone image captured in August 2020, (b) LSI based on the FR model, (c) LSI based on the IV model, (d) LSI based on the CF model, (e) LSI based on the LR model and (f) elevation profile and LSI distribution from the landslide source area to landslide deposit zone.

[Figure]

Figure S1: Spatial characteristics of predicted LSIs: (a) Aerial image acquired in October 2021, (b) LSI based on the FR model, (c) LSI based on the IV model, (d) LSI based on the CF model, (e) LSI based on the LR model and (f) elevation profile and LSI distribution from crown zone to landslide deposit zone.

The varying sensitivity of the FR, IV, CF, and LR models to depositional and source-zone features primarily arises from the inherent differences in their methodological frameworks and ability to capture complex spatial relationships between landslide occurrence and conditioning factors. The LR model, as a multivariate parametric approach, explicitly quantifies the combined and potentially non-linear effects of multiple geomorphological, geological, topographical, hydrological and forest factors, allowing it to capture subtle spatial heterogeneities and interactions that characterize the transition from landslide source zones, marked by steep slopes, high shear stress, and mechanical instability, to depositional zones where slope gradients decrease and sediment accumulates, reducing failure susceptibility. Therefore, LR results in higher predicted susceptibility values in source areas and lower values in depositional zones, reflecting the physical processes of landslide initiation and deposition. In contrast, bivariate heuristic models such as FR, IV, and CF operate on simplified assumptions of factor independence and spatial correlation, evaluating susceptibility based primarily on relative frequency or certainty values within discrete factor classes, which can lead to similar susceptibility assignments across both source and depositional zones if they share common terrain attributes, thereby limiting their spatial discriminatory power.

Consequently, to overcome this issue, we put forth a hybrid integrated strategy to verify whether the LSI derived from the integrated approach aligns with topography, geomorphic features, and landslide characteristics.

The hybrid integrated approach synergistically combines the parametric rigor of LR with the heuristic spatial correlation strengths of FR, IV, and CF models, producing a landslide susceptibility index that aligns closely with the topographic and geomorphic evidence by accurately distinguishing initiation zones of high instability from depositional zones of relative stability. It was observed that the hybrid approach resolves spatial inconsistencies by combining the strengths of multiple models (Fig. 15). For example, a high LSI value was observed in the landslide source area (i.e., 0.75 to 0.9), while a comparatively lower LSI value was observed in the landslide deposit zone (i.e., 0.35-0.55). Each model captures distinct aspects of terrain and susceptibility patterns, and their integration reduces localized prediction errors. Weighted averaging further ensures that models with stronger predictive abilities influence the final susceptibility map more, leading to higher spatial alignment with observed landslide patterns. This cross-temporal validation confirms that the proposed models are not overfit to historical events and can reliably predict landslide-prone zones under varying topographic and geomorphic settings. The hybrid integrated LSI model was further examined using AUC, MSE, MAE, and RMSE with the landslide inventory data exhibiting good consistency with the in-situ observations (AUC=0.908, MSE=0.082, MAE=0.259, and RMSE=0.286). On the other hand, correct classification percentages (for a 0.5 cut-off value) are also calculated to assess the LSI's sensitivity (Gorum et al., 2008). It was exhibited that the integrated models have a prediction capacity of 94.6% (Fig. 17a). The AUC value obtained from the integrated model is 0.909, which also suggests a high landslide prediction rate (Fig. 17b). It is noted that although the anticipated landslide susceptibility index from all the models indicates the degree of field instability, there are variations in their predictive consistency. The LSI predicted based on the hybrid integrated method was consistent with the topographic and landslide characteristics (Bhuyan et al., 2025), suggesting more reliable and appropriate outcomes than other models. This improved spatial consistency is crucial for accurate landslide risk assessment and mitigation. Moreover, the hybrid integrated strategy leverages the complementary strengths of the individual models, yielding improved spatial coherence and practical applicability for real-world landslide hazard management and proactive mitigation planning. Subsequently, the LSI calculated through the hybrid integrated approach was used

for further analysis.

[Figure]

Figure 15. LSI based on the hybrid integrated approach: (a) spatial distribution of LSI in the Jecheon-si region, (b-d) the details of LSI distribution of three recent past landslide events, and (b'-d') elevation profile and LSI distribution from the landslide source area to the deposit zone at different landslide sites.

[Figure]

Figure 17. (a) the estimated LSI corresponding to the landslide inventory datasets with correct classification percentages, and (b) model performance of the proposed integrated approach based on the AUC.

**Comment 3:** There is also a lack of in-depth analysis of model limitations, data uncertainties, and regional applicability, which significantly weakens the scientific value of the work. Based on these issues, I recommend rejection.

**Response:** We appreciate the reviewer's concern regarding the limitations of the applied FR, IV, CF, and LR models, as well as data uncertainties, and the regional applicability of the proposed hybrid integrated approach. In the revision, we have expanded the discussion on the sensitivity of the LSI models and examined how these widely used approaches may influence susceptibility mapping results. We have also explicitly addressed data-related uncertainties, including inventory completeness and potential biases in landslide sampling. Furthermore, we evaluated the regional applicability of the proposed approach by discussing its transferability to other regions with similar geomorphological and climatic settings, along with the associated constraints. These additions aim to enhance the scientific value of the present work by providing a transparent and critical assessment of methodological and contextual limitations. Accordingly, we updated several sections by incorporating existing model limitations, data uncertainties, limitations of the present study, and regional applicability of the proposed model, as highlighted below.

Landslide susceptibility models, whether statistical, probabilistic, or machine-learning based, inevitably face limitations due to the complex, non-linear, and site-specific nature of landslide processes, the heterogeneity of conditioning factors, and the variability of triggering mechanisms across temporal and spatial scales. While numerous statistical approaches such as FR, IV, CF, and LR have been successfully applied on various geographic regions (Merghadi et al., 2020; Shano et al., 2020; Park et al., 2013), none are universally optimal, and their predictive capacity depends heavily on inventory quality, factor relevance, and geomorphological context (Aditian et al., 2018; Tang et al., 2020). Furthermore, a landslide susceptibility index typically indicates areas that are more prone to landslides based on various factors and parameters. Thus, previous studies have primarily focused on assessing the overall performance of predicted susceptibility rather than examining the spatial characteristics of the predicted LSI. The overall accuracy of widely accepted models may produce acceptable LSIs in terms of AUC, MAE and RMSE, but it may not always be comparable with the topographic, geomorphic and landslide characteristics. Therefore, the main novelties of the present investigation include the development of LSI models using different widely adopted statistical models and uncertainty evaluation based on the spatial characteristics of the predicted landslide

susceptibility index to study previously overlooked accuracy criteria and propose a hybrid integrated approach to achieve higher accuracy than the individual LSI models. We also used both AUC-based performance metrics and error-based measures (MAE, MSE, RMSE) for each LSI model, comparing training and testing datasets. This dual evaluation mitigates the over-reliance on AUC alone and provides a more robust understanding of prediction quality. We also performed multicollinearity diagnostics to ensure that selected variables met VIF and tolerance thresholds, reducing redundancy-induced uncertainty. Subsequently, we have highlighted spatial consistency analysis of the predicted LSIs as an innovative aspect of this study. By examining the susceptibility patterns in both source zones and deposit zones, we identified discrepancies where high AUC scores did not necessarily align with known landslide initiation areas. This analysis provides additional diagnostic insight into model robustness, beyond conventional accuracy statistics. To overcome this issue, we put forth a hybrid integrated strategy to verify whether the LSI derived from the integrated approach aligns with topography, geomorphic features, and landslide characteristics. It was observed that the LSI predicted through the integrated approach resolves spatial inconsistencies by combining the strengths of multiple models, which was not in the earlier case. For example, a high LSI value was observed in the landslide source area (i.e., 0.75 to 0.9), while a comparatively lower LSI value was observed in the landslide deposit zone (i.e., 0.35-0.55). This cross-temporal validation confirms that the proposed models are not overfit to historical events and can reliably predict landslide-prone zones under varying topographic and geomorphic settings. Moreover, the hybrid integrated strategy leverages the complementary strengths of the individual models, yielding improved spatial coherence and practical applicability for real-world landslide hazard management and proactive mitigation planning.

Although the developed hybrid integrated LSI model and LHM map revealed acceptable prediction accuracy with spatial consistency, this approach has several inherent limitations due to the complex, non-linear nature of landslide processes, the heterogeneity of conditioning factors, and the spatio-temporal variability of triggering mechanisms. In the present study, the landslide inventory (n=160) was compiled from multi-sourced data, including aerial photographs, historical Google Earth imagery, field investigations, and recorded data from the Korea Forest Service (KFS). Although diverse in origin, this inventory is spatially clustered in the central to northern part of the study region, reflecting spatial rainfall patterns controls. The moderate inventory size and spatial clustering limit the events-pervariable ratio in LSI models and risk of pseudo-replication, increasing parameter uncertainty and reducing generalization capability. Moreover, the class imbalance may bias model calibration toward the dominant class. Further, for LSI model development, we considered 18 influencing factors, viz., topographic slope, aspect, landforms class, average shear-wave velocity, TPI, CI, TWI, TRI, plan curvature, profile curvature, SPI, SL, surface lithology, soil thickness, timber density, timber age, soil type, and timber diameter. These factors are widely used for the LSI models in South Korea; however, the inclusion of dynamic influencing factors such as high-resolution NDVI and LULC in future studies may increase spatial consistency by incorporating characteristics of landslide dynamics in the mountainous region, as frequent forest fires impact the South Korean mountainous region. Further, we conducted multicollinearity diagnostics (Table 2) to ensure statistical robustness of the selected influencing factors. However, additional collinearity and feature-importance tests, such as the pearson correlation coefficient (PCC) and information gain ratio, could be applied to refine the selection of influencing factors. Moreover, a machine learning-based feature selection approach may be adopted in future studies to optimize model performance further.

In addition, the applied FR, IV, CF, LR, and integrated hybrid models rely on the premise that future landslides occur under conditions similar to past events; however, this assumption may lead to model overfitting, limited generalization, or bias if the historical inventory is incomplete or unrepresentative. It is also acknowledged that even though FR, IV, and CF models differ in their computation and interpretation of spatial relationships between landslide events and conditioning factors, they share similar probabilistic foundations. Therefore, incorporating advanced ML and ensemble algorithms in future modeling could further enhance the accuracy and predictive power of LSIs. Further, the inclusion of maximum daily rainfall intensity (2000-2019) as a triggering factor to determine LHM of the region enhances the model's temporal relevance, yet regional rainfall thresholds vary and require site-specific adjustment. Therefore, in future studies, we intend to use site-specific long-term rainfall intensity data from AWSs to better account for local variations in triggering factors, which could improve the landslide hazard microzonation zones. Further, the LSI models utilized in the present study were calibrated for the humid monsoon climate, steep terrain, and lithological complexity of Jecheon-si region, South Korea. While the methodological framework is transferable to other regions, the factor weights and model coefficients should be recalibrated based on the local inventories and environmental conditions.

**References**

Aditian, A., Kubota, T., and Shinohara, Y.: Comparison of GIS-based landslide susceptibility models using frequency ratio, logistic regression, and artificial neural network in a tertiary region of Ambon, Indonesia, Geomorphology, 318, 101-111, https://doi.org/10.1016/j.geomorph.2018.06.006, 2018.

Bhuyan, K., Rana, K., Ozturk, U., Nava, L., Rosi, A., Meena, S.R., Fan, X., Floris, M., van Westen, C. and Catani, F.: Towards automatic delineation of landslide source and runout, Eng. Geol., 345, 107866, https://doi.org/10.1016/j.enggeo.2024.107866, 2025.

Ding, D., Wu, Y., Wu, T., and Gong, C.: Landslide susceptibility assessment in Tongguan District Anhui China using information value and certainty factor models, Sci. Rep., 15(1), 12275, https://doi.org/10.1038/s41598-025-93704-z, 2025.

Gorum, T., Gonencgil, B., Gokceoglu, C., and Nefeslioglu, H. A.: Implementation of reconstructed geomorphologic units in landslide susceptibility mapping: the Melen Gorge (NW Turkey), Nat. Hazards, 46(3), 323-351, https://doi.org/10.1007/s11069-007-9190-6, 2008.

He, Y., Zhao, Z. A., Yang, W., Yan, H., Wang, W., Yao, S., Zhang, L., and Liu, T.: A unified network of information considering superimposed landslide factors sequence and pixel spatial neighbourhood for landslide susceptibility mapping, Int. J. Appl. Earth Obs. Geoinf., 104, 102508, https://doi.org/10.1016/ j.jag. 2021.102508, 2021.

Merghadi, A., Yunus, A. P., Dou, J., Whiteley, J., ThaiPham, B., Bui, D. T., Avtar, R., and Abderrahmane, B.: Machine learning methods for landslide susceptibility studies: A comparative overview of algorithm performance, Earth Sci. Rev., 207, 103225, https://doi.org/10.1016/j.earscirev.2020.103225, 2020.

Nguyen, Q.H., Ly, H.B., Ho, L.S., Al-Ansari, N., Le, H.V., Tran, V.Q., Prakash, I. and Pham, B.T.: Influence of data splitting on performance of machine learning models in prediction of shear strength of soil, Math. Prob. Eng., 2021(1), 4832864, https://doi.org/10.1155/2021/4832864, 2021.

Park, S., Choi, C., Kim, B., and Kim, J.: Landslide susceptibility mapping using frequency ratio, analytic hierarchy process, logistic regression, and artificial neural network methods at the Inje area, Korea. Environ. Earth Sci. 68(5), 1443-1464, https://doi.org/10.1007/s12665-012-1842-5, 2013.

Shano, L., Raghuvanshi, T. K., and Meten, M.: Landslide susceptibility evaluation and hazard zonation techniques–a review, Geoenviron. Disasters, 7(1), 1-19, https://doi.org/10.1186/s40677-020-00152-0, 2020.

Tang, Y., Feng, F., Guo, Z., Feng, W., Li, Z., Wang, J., Sun, Q., Ma, H. and Li, Y.: Integrating principal component analysis with statistically-based models for analysis of causal factors and landslide susceptibility mapping: A comparative study from the loess plateau area in Shanxi (China), J. Clean. Prod., 277, 124159, https://doi.org/10.1016/j.jclepro.2020.124159, 2020.

Xie, Z., Chen, G., Meng, X., Zhang, Y., Qiao, L., and Tan, L.: A comparative study of landslide

susceptibility mapping using weight of evidence, logistic regression and support vector machine and evaluated by SBAS-InSAR monitoring: Zhouqu to Wudu segment in Bailong River Basin, China. Environ. Earth Sci., 76(8), 313, https://doi.org/10.1007/s12665-017-6640-7, 2017.

Zhou, X., Wu, W., Qin, Y., and Fu, X.: Geoinformation-based landslide susceptibility mapping in subtropical area, Sci. Rep., 11(1), 1-16, https://doi.org/10.1038/ s41598-021-03743-5, 2021b.

---

## Author Comment (AC2)

**Manuscript ID: egusphere-2025-1169**

My co-authors and I would like to express our gratitude to the reviewers for their constructive feedback and suggestions for strengthening our research. The changes we have made to the attached file in response to such feedback and suggestions have been highlighted in blue to facilitate their identification. I would also like to offer my apologies for the length of time it took us to prepare this response.

**Response to Reviewer #2**

We greatly appreciate the critical review of the manuscript and the constructive suggestions put forth by the reviewer that will help improve the quality of the manuscript. We have responded point by point to all the comments and suggestions raised by Reviwer#2 as follows:

**Comment 1:** The manuscript has some serious problems in methodology and data. For the methodology, the used model such as frequency ratio (FR), certainty factor (CF), logistic regression (LR), and information value (IV), is not novel and new. Moreover, the as frequency ratio (FR), certainty factor (CF), and information value (IV) is similar and almost same models based on probability. I recommend the new models such as machine learning models.

**Response:** We appreciate the reviewer's comment and acknowledge the increasing prominence of advanced machine learning (ML) methodologies in landslide susceptibility mapping due to their ability to handle complex datasets and improve predictive performance. However, our choice of statistical methods (FR, IV, CF, and LR) was intentional and grounded in both methodological and practical considerations. Although FR, IV, and CF share probabilistic foundations, they differ in computation and in the interpretation of spatial relationships between landslide events and conditioning factors. These models, along with LR, have been widely used for LSI mapping and validated across various regions (e.g., Lee and Pradhan, 2007; Aditian et al., 2018; Dash et al., 2022), allowing for meaningful comparative benchmarking in our study area.

Furthermore, the spatial distribution of rainfall-induced landslide events in our study area is highly clustered, particularly in the central and northern regions of Jecheon-si. This pattern corresponds closely with the distribution of maximum daily rainfall intensity (Fig. 5), indicating that intense rainfall is a dominant landslide-triggering factor. The compiled landslide inventory is both limited and spatially uneven, making it less suitable for training data-intensive

ML models. Therefore, in such data-constrained conditions, interpretable and efficient statistical models remain practical and reliable tools for landslide susceptibility analysis (Ding et al., 2025; Xie et al., 2017). Besides, the primary objective of this study is to evaluate the predictive performance and spatial consistency of commonly used statistical models and to propose a hybrid integration approach that leverages their combined strengths. Our methodology emphasizes spatial validation and model synergy rather than novelty in algorithm selection. Accordingly, in the revision, we have highlighted both the limitations of existing LSI models and the novel contributions of the present investigation in the Introduction section, as given below,

In the past two decades, several statistical and machine-learning approaches have been introduced for landslide susceptibility assessment, presuming that landslides trigger in a similar environment to prior landslides (Xing et al., 2021; Aditian et al., 2018; Park et al., 2013; Lee and Pradhan, 2007; Lee et al., 2002). Although numerous techniques have been put forth to create GIS-based landslide susceptibility maps, there still needs to be an agreement on the best practices (Tang et al., 2020; Aditian et al., 2018). Most quantitative methods considered past landslides to determine the ranks and weight of each factor attribute based on their spatial association. Consequently, numerous quantitative have been frequently applied for landslide susceptibility mapping, ranging from conventional statistical methods (e.g., frequency ratio, information value, Shannon entropy, certainty factor, weights of evidence, and logistic regression) to advanced machine learning algorithms (e.g., random forests, support vector machines, extreme gradient boosting, and neural networks) (Biswas et al., 2023; Dash et al., 2022; Mandal et al., 2021; Zhou et al., 2021a; Pham et al., 2020; Riaz et al., 2018; Aditian et al., 2018; Shahabi and Hashim, 2015; Park et al., 2013). The advantages and limitations of these statistical and probabilistic models have been systematically reviewed by Merghadi et al. (2020) and Shano et al. (2020). Even though there were numerous studies on landslide susceptibility, no single approach is suitable for all cases. As a result, to determine landslide susceptibility in a given area, the best model must be chosen based on the landslide's characteristics and the accessibility of inventory data (Zhu et al., 2018). Consequently, it is still crucial to calculate the effectiveness of various models for particular landslide susceptibility procedures. In addition, model integration provides another opportunity to improve model accuracy by combining different models on the GIS platform (Barman et al., 2024).

A landslide susceptibility index typically indicates areas that are more prone to

landslides based on various factors and parameters. Thus, previous studies have primarily focused on assessing the overall performance of predicted susceptibility rather than examining the predicted LSI's spatial geomorphic and topographic characteristics. The overall accuracy of widely accepted models may produce acceptable LSIs in terms of AUC, MAE, MSE, and RMSE, but it may not always be comparable with the landslide characteristics. For example, the landslide source area (or depletion zones), characterized by steep slopes, high topographic relief, and often exposed bedrock or weathered material, is critical in understanding landslide dynamics, as it is where the initial failure occurs, leading to material movement (Crosta et al., 2003). Thus, this area is typically characterized by high susceptibility to landslides due to the inherent instability (Bhuyan et al., 2025; Mathews et al., 2024). Identifying these areas is crucial for understanding initiation mechanisms and improving hazard models. Conversely, the landslide deposit area, characterized by gentle slopes and often covered by accumulated landslide debris, is the final resting place of the landslide mass and generally exhibits lower susceptibility due to the stabilization of the material (Meyrat et al., 2022). Numerous studies show that the characteristics of the deposit area can influence future landslide risks, as loose materials remain vulnerable (Li et al., 2024). Therefore, the landslide characteristics along with the AUC, MAE, MSE, and RMSE should be analyzed to validate the predicted LSI values. However, most landslide studies consider the overall model performance (i.e., AUC) and ignore the spatial inconsistency phenomenon. Therefore, the main novelties of this study include (a) the development of landslide susceptibility (LSI) maps by comparing and analyzing different statistical models commonly used for assessing LSI, (b) LSI models were validated using AUC and other statistical methods, (c) evaluating spatial geoporphic and topographic characteristics of the predicted LSIs to study previously overlooked accuracy criteria, (c) proposed a hybrid integrated approach to achieve higher accuracy than the individual LSI models, and (d) prepared a reliable landslide hazard microzonation map through integration of triggering factor (i.e., maximum rainfall intensity (mm/day) from 2000 to 2019) to mitigate landslide-induced disaster risks appropriately.

However, we agree that integrating ML algorithms can enhance the accuracy and predictive power of LSIs. Consiquently, in revision, we have acknowledged this limitation in the discussion section.

**Comment 2:** Also, the relationships between landslide and input factors should be analyzed for the reason using the FR. Then only related factors should be used for the analysis. The authors just describe the result of FR and used all factors which is related to landslide or not.

**Response:** Thank you for your insightful observations. In the present study, landslide susceptibility was modeled using four statistical models: FR, IV, CF, and LR. Five major steps were followed to achieve this goal: (a) the spatial relationship between the predisposing factors and landslide inventory were analyses based on the FR values, (b) Multicollinearity analysis of predisposing factors to understanding collinearity among the predisposing factors and select suitable predisposing factors for the analysis, (c) a GIS-based normalized raster database of 18 predisposing factors were prepared to calculate the FR, IV, and CF values and to perform subsequent analysis, (d) LR analysis was performed based on the dependent (landslide inventory data) and independent variables, and (e) calculated LSIs were validated using AUC and other statistical methods. To train and validate the LSI models, we randomly split the landslide inventory dataset into training and testing, i.e., 70% and 30%, respectively (Nguyen et al., 2021; Zhou et al., 2021b; Aditian et al., 2018). This study aims to compare the most widely used landslide susceptibility approaches and gain insight into their precision in predicting capacities in susceptible zones.

As the reviewer noted, we performed FR analysis to assess the degree of association between each factor's attributes and historical landslide events. This was discussed in Section 4.1 of the revised manuscript, where we emphasize that higher FR values indicate a stronger correlation between a given factor class and landslide occurrence (Lee and Talib, 2005). The FR results helped us interpret the influence of each factor on landslide distribution. Additionally, we conducted multicollinearity diagnostics to ensure statistical robustness in the LSI model. All 18 predisposing factors satisfied the recommended thresholds (Variance Inflation Factor < 10 and Tolerance > 0.1) as per Zhang et al. (2020) and Kadavi et al. (2019). For instance, the Topographic Roughness Index (TRI) exhibited the highest VIF (6.483) and lowest tolerance (0.154), both well within acceptable limits. Although some factor classes may exhibit weaker FR values, we retained all 18 factors due to consistency across all four models (FR, IV, CF, LR) to allow comparative evaluation. We have clarified this rationale in the revised manuscript and highlighted that future studies may adopt machine learning-based feature selection to further optimize model performance.

**Comment 3:** For the data, the authors have used the 112 landslides The number of landslides is too small to apply the modes. The Also, the study does not explicitly describe using a separate validation dataset (e.g., splitting the 112 landslides into training/testing subsets). Performance metrics like AUC were apparently computed on the same inventory used for modeling, which could lead to overfitting concerns.

**Response:** Thank you for your insightful observation. In the present study, the inventory database was created using high-resolution aerial imagery (obtained from http://map.ngii.go.kr/ms/map/), historical Google Earth imagery, field investigation and compilation of available inventory data from the Korea Forest Service (KFS). As suggested, we further rigorously used historical Google Earth imagery and aerial photographs to update the landslide inventory data. Subsequently, we prepared an updated landslide inventory ($n$=160) map of the Jecheon-si region, as shown in Fig. 5. Additional, in the revised manuscript, to better understand the spatial distribution of landslide events in this region, we analyzed the long-term (2000-2019) maximum daily rainfall intensities using TRMM (Tropical Rainfall Measuring Mission, https://gpm.nasa.gov/missions/trmm) datasets processed in Google Earth Engine. Results revealed that maximum rainfall intensity ranged from 43.74 to 56.7 mm/day across the region, with the central and northern parts exhibiting significantly higher values (Fig. 5). This pattern aligns with the concentration of landslide events in the northern and central parts of the study region, indicating higher rainfall intensity is a critical factor controlling the spatial distribution of landslides. Subsequently, long-term rainfall intensity was utilized as a triggering factor for the landslide hazard microzonation analysis.

[Figure]

Figure 5. Maximum rainfall intensity (mm/day) distribution of the region for the period of 2000 to 2019, and spatial distribution of landslide inventory.

To train and validate the LSI models, we randomly split the landslide inventory dataset into training and testing, i.e., 70% and 30%, respectively (Barman et a., 2023; Nguyen et al., 2021; Zhou et al., 2021b; Aditian et al., 2018). Figure 13 illustrates the ROC-AUC values of the FR, IV, CF and LR models, which reveal that all four models demonstrate strong discriminatory power, with AUC values consistently exceeding 0.85, indicating high model reliability. It was observed that the LR model achieved the highest AUC values for both training (0.901) and testing (0.930) datasets, signifying its superior generalization capability and robustness in capturing the non-linear relationships between landslide occurrence and contributing factors. The FR model yielded AUCs of 0.879 (training) and 0.905 (testing), followed closely by IV (0.858 and 0.910) and CF (0.861 and 0.917), reflecting their competency in bivariate and heuristic-based spatial correlation assessments. Nevertheless, all models yielded acceptable and comparable prediction accuracies in both the training and testing datasets, indicating their robustness for landslide susceptibility mapping without clear

signs of overfitting.

[Figure]

Figure 13. ROC curves and corresponding AUC values for four LSI models: (a) LR, (b) FR, (c) IV and (d) CF models [Note: Blue and red curves denote training and testing datasets, respectively, used for model evaluation. The block dotted line represents the random guess].

**Comment 3:** A randomized train-test split or cross-validation would increase confidence that the models generalize beyond the known inventory.

**Response:** Thank you for your comment. In the present study, to train and validate the LSI models, the landslide inventory dataset was randomly split into 70% for training and 30% for testing (Nguyen et al., 2021; Zhou et al., 2021b; Aditian et al., 2018). Consequently, FR, IV, CF and LR models were developed based on the 70% training data. Figure 13 illustrates the ROC-AUC values of the FR, IV, CF and LR models, which reveal that all four models demonstrate strong discriminatory power, with AUC values consistently exceeding 0.85, indicating high model reliability. It was observed that the LR model achieved the highest AUC values for both training (0.901) and testing (0.930) datasets, signifying its superior generalization capability and robustness in capturing the non-linear relationships between landslide occurrence and contributing factors. The FR model yielded AUCs of 0.879 (training)

and 0.905 (testing), followed closely by IV (0.858 and 0.910) and CF (0.861 and 0.917), reflecting their competency in bivariate and heuristic-based spatial correlation assessments. Further, based on the training and testing datasets, the models' accuracy was examined using RMSE, MSE, and MAE. The outcome demonstrates that the FR, CF and IV models had the lowest RMSE, MSE, and MAE values (Table 4). On the other hand, the LR model had higher MAE, MSE, and RMSE values and comparatively lower prediction accuracy than other models. Nevertheless, all models yielded acceptable and comparable prediction accuracies in both the training and testing datasets, indicating their robustness for landslide susceptibility mapping. Therefore, selecting an appropriate model for landslide susceptibility mapping is difficult, even though the performances and prediction accuracy of all the discussed models were acceptable.

[Figure]

Figure 13. ROC curves and corresponding AUC values for four LSI models: (a) LR, (b) FR, (c) IV and (d) CF models [Note: Blue and red curves denote training and testing datasets, respectively, used for model evaluation. The block dotted line represents the random guess].

Table 4. Validation of models by AUC, RMSE, MSE, and MAE.

| Models | Traing | | | | Validation | | | |
|---|---|---|---|---|---|---|---|---|
| | AUC | MAE | MSE | RMSE | AUC | MAE | MSE | RMSE |
| FR | 0.879 | 0.291 | 0.092 | 0.303 | 0.905 | 0.267 | 0.078 | 0.279 |
| IV | 0.858 | 0.248 | 0.068 | 0.261 | 0.910 | 0.220 | 0.053 | 0.230 |
| CF | 0.861 | 0.236 | 0.063 | 0.252 | 0.917 | 0.204 | 0.046 | 0.215 |
| LR | 0.901 | 0.306 | 0.174 | 0.417 | 0.930 | 0.238 | 0.125 | 0.354 |

On the other hand, we utilized the entire inventory database to validate the integrated LSI model and landslide hazard microzonation map along with in-situ observations. The hybrid integrated LSI model was examined using AUC, MSE, MAE, and RMSE, with the landslide inventory data exhibiting good consistency with the in-situ observations. On the other hand, correct classification percentages (for 0.5 cut-off value) are also calculated to assess the LSI's sensitivity (Gorum et al., 2008). It was exhibited that the integrated models have a prediction capacity of 95% (Fig. 17a). The AUC value obtained from the integrated model is 0.908, which also suggests a high landslide prediction rate (Fig. 17b).

[Figure]

Figure 17. (a) The estimated LSI corresponding to the landslide inventory datasets with correct classification percentages, and (b) Model performance of the proposed integrated approach based on the AUC.

The above has been incorporated into the revised manuscript in Sections 4 and 5.

**Comment 4:** There is no "Discussion section. The section should be included with "Conclusion" section.

**Response:** Thank you for your comment. As recommended, the discussion section have been incorporated in the revised manuscript.

**5. Discussion**

Landslides, driven by downslope movements of soil and rock, are increasingly frequent in South Korea due to heavy monsoon rainfall and climate change. Urbanization and deforestation are anticipated to intensify these events, underscoring the need for effective hazard zonation mapping to mitigate risks and safeguard the well-being of communities and infrastructure. This study developed a hybrid integrated approach to improve the accuracy and spatial consistency of anticipated LSIs. By combining the strengths of multiple widely accepted statistical models (FR, CF, IV, and LR), the hybrid integrated approach effectively addresses the limitations of individual models. For these purposes, we comprehensively evaluate the predicted LSI values derived from the FR, CF, IV, and LR models and the one proposed hybrid model through traditional accuracy metrics (AUC, RMSE, MSE, MAE) in conjunction with with spatial topographic and landslide characteristies, focused on landslide source areas (steep, unstable zones) and deposition areas (stabilized material zones), which reflect heterogeneities in mechanical instability (Crosta et al., 2003; Mathews et al., 2024; Bhuyan et al., 2025; Meyrat et al., 2022; Li et al., 2024).

The LSI models (i.e., FR, CF, IV, and LR) were developed using landslide inventories and 18 widely used influencing factors, i.e., topography, hydrogeology, soils, forests, and lithology. Subsequently, we performed FR analysis to assess the degree of association between each factor's attributes and historical landslide events. We diagnosed the predisposing factors through multicollinearity tests and found no collinearity among the independent predisposing factors. After that, LSI models were developed based on the eighteen variables (Fig. 12, in the revised manuscript). Based on the training and testing datasets, the models' accuracy was examined using AUC, RMSE, MSE, and MAE. The outcome demonstrates that the FR, CF and IV models had the lowest RMSE, MSE, and MAE values (Table 4). On the other hand, the LR model had higher MAE, MSE, and RMSE values and comparatively lower prediction accuracy than other models.

Table 4. Validation of models by AUC, RMSE, MSE, and MAE.

| Models | Traing | | | | Validation | | | |
|---|---|---|---|---|---|---|---|---|
| | AUC | MAE | MSE | RMSE | AUC | MAE | MSE | RMSE |
| FR | 0.879 | 0.291 | 0.092 | 0.303 | 0.905 | 0.267 | 0.078 | 0.279 |
| IV | 0.858 | 0.248 | 0.068 | 0.261 | 0.910 | 0.220 | 0.053 | 0.230 |
| CF | 0.861 | 0.236 | 0.063 | 0.252 | 0.917 | 0.204 | 0.046 | 0.215 |
| LR | 0.901 | 0.306 | 0.174 | 0.417 | 0.930 | 0.238 | 0.125 | 0.354 |

Further, it was observed that all models achieved AUC > 0.85 for both training and testing datasets, indicating high discriminative ability. The LR model yielded the highest AUC (0.901 training, 0.930 testing), while FR, IV, and CF achieved slightly lower but comparable results (0.858-0.910). Although the LR model exhibits slightly higher MAE, MSE, and RMSE values, all models demonstrated robust predictive accuracy without clear signs of overfitting. Thereafter, to explicitly assess model generalization in real-world scenarios, we tested spatial characteristics of the model outputs against recent landslide sites not included in the inventory by utilizing high-resolution LiDAR DEM, aerial photographs (NGII, 2020-2021), and drone surveys (August 2020). It was observed that even though the statistical model evaluation revealed high AUC values and acceptable statistical metrics, spatial analysis revealed inconsistencies of LSI distribution in landslide source and deposition zones (Figs. 14 and S1). For instance, FR, IV, and CF models predicted high LSI values in both the crown (~0.75) and deposit (~0.69) zones, whereas LR predicted moderate susceptibility in the crown (~0.62) but very low in the deposit (~0.05). The four applied models were found to be able to locate the landslide source area precisely; however, they are not consistent with the landslide and topography characteristics. On the other hand, the spatial analysis of predicted LSI values reveals that the hybrid integrated approach better captures the topographic and geomorphic characteristics of landslide source and deposition areas compared to individual models (Fig. 15). The hybrid integrated LSI model was further examined using AUC, MSE, MAE, and RMSE with the landslide inventory data exhibiting good consistency with the in-situ observations (AUC=0.908, MSE=0.082, MAE=0.259, and RMSE=0.286). On the other hand, correct classification percentages (for 0.5 cut-off value) are also calculated to assess the LSI's sensitivity (Gorum et al., 2008). It was exhibited that the integrated models have a prediction capacity of 95% (Fig. 17a). The AUC value obtained from the integrated model is 0.908, which also suggests a high landslide prediction rate (Fig. 17b). It is noted that although the anticipated

landslide susceptibility index from all the models indicates the degree of field instability, there are variations in their predictive consistency. The LSI predicted based on the hybrid integrated method was consistent with the topographic and landslide characteristics (Bhuyan et al., 2025), suggesting more reliable and appropriate outcomes than other models. This improved spatial consistency is crucial for accurate landslide risk assessment and mitigation.

[Figure]

Figure 17. (a) the estimated LSI corresponding to the landslide inventory datasets with correct classification percentages, and (b) model performance of the proposed integrated approach based on the AUC.

Thereafter, an LHM map was developed by integrating the optimized hybrid LSI with maximum rainfall intensity (2000-2019), which was identified as a dominant landslide-triggering factor in this region. The analysis revealed that severe hazard zones cover approximately 77.17 $km^2$, predominantly covered in the northern part of the study area (Fig. 16). This finding is supported by TRMM-derived long-term maximum rainfall intensity patterns, which show a strong correlation between rainfall and landslide occurrences in these areas. Previous studies (Lee et al., 2020; Park and Lee, 2021) similarly highlighted that 95% of landslides in South Korea occur during the monsoon season. The LHM map, validated against field observations, revealed that 95% of landslide occurrences align with severe to high-hazard zones. The precision results (i.e., R-index) also indicate that the developed LHM map has a very high prediction accuracy and is useful for landcover planning and landslide-induced disaster mitigation purposes.

Although the developed hybrid integrated LSI model and LHM map revealed acceptable prediction accuracy with spatial consistency, this approach has several inherent limitations due to the complex, non-linear nature of landslide processes, the heterogeneity of conditioning factors, and the spatio-temporal variability of triggering mechanisms. In the present study, the landslide inventory (n=160) was compiled from multi-sourced data, including aerial photographs, historical Google Earth imagery, field investigations, and recorded data from the Korea Forest Service (KFS). Although diverse in origin, this inventory is spatially clustered in the central to northern part of the study region, reflecting spatial rainfall patterns controls. The moderate inventory size and spatial clustering limit the events-per-variable ratio in LSI models and risk of pseudo-replication, increasing parameter uncertainty and reducing generalization capability. Moreover, the class imbalance may bias model calibration toward the dominant class. Further, for LSI model development, we considered 18 influencing factors, viz., topographic slope, aspect, landforms class, average shear-wave velocity, TPI, CI, TWI, TRI, plan curvature, profile curvature, SPI, SL, surface lithology, soil thickness, timber density, timber age, soil type, and timber diameter. These factors are widely used for the LSI models in South Korea; however, the inclusion of dynamic influencing factors such as high-resolution NDVI and LULC in future studies may increase spatial consistency by incorporating characteristics of landslide dynamics in the mountainous region, as frequent forest fires impact the South Korean mountainous region. Further, we conducted multicollinearity diagnostics (Table 2) to ensure statistical robustness of the selected influencing factors. However, additional collinearity and feature-importance tests, such as the pearson correlation coefficient (PCC) and information gain ratio, could be applied to refine the selection of influencing factors. Moreover, a machine learning-based feature selection approach may be adopted in future studies to optimize model performance further.

In addition, the applied FR, IV, CF, LR, and integrated hybrid models rely on the premise that future landslides occur under conditions similar to past events; however, this assumption may lead to model overfitting, limited generalization, or bias if the historical inventory is incomplete or unrepresentative. It is also acknowledged that even though FR, IV, and CF models differ in their computation and interpretation of spatial relationships between landslide events and conditioning factors, they share similar probabilistic foundations. Therefore, incorporating advanced ML and ensemble algorithms in future modeling could further enhance the accuracy and predictive power of LSIs. Further, the inclusion of maximum

daily rainfall intensity (2000-2019) as a triggering factor to determine LHM of the region enhances the model's temporal relevance, yet regional rainfall thresholds vary and require site-specific adjustment. Therefore, in future studies, we intend to use site-specific long-term rainfall intensity data from AWSs to better account for local variations in triggering factors, which could improve the landslide hazard microzonation zones. Further, the LSI models utilized in the present study were calibrated for the humid monsoon climate, steep terrain, and lithological complexity of Jecheon-si region, South Korea. While the methodological framework is transferable to other regions, the factor weights and model coefficients should be recalibrated based on the local inventories and environmental conditions.

**References**

Aditian, A., Kubota, T., and Shinohara, Y.: Comparison of GIS-based landslide susceptibility models using frequency ratio, logistic regression, and artificial neural network in a tertiary region of Ambon, Indonesia, Geomorphology, 318, 101-111, https://doi.org/10.1016/j.geomorph.2018.06.006, 2018.

Barman, J., Biswas, B., and Rao, K. S.: A hybrid integration of analytical hierarchy process (AHP) and the multiobjective optimization on the basis of ratio analysis (MOORA) for landslide susceptibility zonation of Aizawl, India, Nat. Hazards, 120, 8571-8596, https://doi.org/10.1007/s11069-024-06538-9, 2024.

Bhuyan, K., Rana, K., Ozturk, U., Nava, L., Rosi, A., Meena, S.R., Fan, X., Floris, M., van Westen, C. and Catani, F.: Towards automatic delineation of landslide source and runout, Eng. Geol., 345, 107866, https://doi.org/10.1016/j.enggeo.2024.107866, 2025.

Biswas, B., Rahaman, A., and Barman, J.: Comparative assessment of FR and AHP models for landslide susceptibility mapping for Sikkim, India and preparation of suitable mitigation techniques, J. Geol. Soc. India, 99(6), 791-801, https://doi.org/10.1007/s12594-023-2386-x, 2023.

Crosta, G., Imposimato, S., and Roddeman, D.: Numerical modelling of large landslides stability and runout, Nat. Hazards Earth Syst. Sci., 3(6), 523–538, https://doi.org/10.5194/nhess-3-523-2003, 2003.

Dash, R. K., Falae, P. O., and Kanungo, D. P.: Debris flow susceptibility zonation using statistical models in parts of Northwest Indian Himalayas-implementation, validation, and comparative evaluation, Nat. Hazards, 111(2), 2011-2058, https://doi.org/10.1007/s11069-021-05128-3, 2022.

Ding, D., Wu, Y., Wu, T., and Gong, C.: Landslide susceptibility assessment in Tongguan District Anhui China using information value and certainty factor models, Sci. Rep., 15(1), 12275, https://doi.org/10.1038/s41598-025-93704-z, 2025.

Kadavi, P. R., Lee, C. W., and Lee, S.: Landslide-susceptibility mapping in Gangwon-do, South Korea, using logistic regression and decision tree models, Environ. Earth Sci., 78(4), 1-

17, https://doi.org/10.1007/s12665-019-8119-1, 2019.

Lee, S., and Talib, J. A.: Probabilistic landslide susceptibility and factor effect analysis, Environ. Geol., 47(7), 982-990, https://doi.org/10.1007/s00254-005-1228-z, 2005.

Lee, S., and Pradhan, B.: Landslide hazard mapping at Selangor, Malaysia using frequency ratio and logistic regression models, Landslides, 4, 33–41, https://doi.org/10.1007/s10346-006-0047-y , 2007.

Lee, S., Chwae, U., and Min, K.: Landslide susceptibility mapping by correlation between topography and geological structure: the Janghung area, Korea, Geomorphology, 46(3-4), 149-162, https://doi.org/10.1016/S0169-555X(02)00057-0, 2002.

Lee, D. H., Kim, Y. T., and Lee, S. R.: Shallow landslide susceptibility models based on artificial neural networks considering the factor selection method and various non-linear activation functions, Remote Sens., 12(7), 1194, https://doi.org/10.3390/rs12071194, 2020.

Li, W. P., Wu, Y. M., Gao, X., Wang, W. M., Yang, Z. H., and Liu, H. J.: The Distribution Pattern of Ground Movement and Co-Seismic Landslides: A Case Study of the 5 September 2022 Luding Earthquake, China, J. Geophys. Res. Earth Surf., 129(5), e2023JF007534, https://doi.org/10.1029/2023JF007534, 2024.

Mandal, K., Saha, S., and Mandal, S.: Applying deep learning and benchmark machine learning algorithms for landslide susceptibility modelling in Rorachu river basin of Sikkim Himalaya, India, Geosci. Front., 12(5), 10120, https://doi.org/ 10.1016/j.gsf.2021.101203, 2021.

Mathews, N. W., Leshchinsky, B. A., Mirus, B. B., Olsen, M. J., and Booth, A. M. RegionGrow3D: A deterministic analysis for characterizing discrete three-dimensional landslide source areas on a regional scale, J. Geophys. Res. Earth Surf., 129(9), e2024JF007815, https://doi.org/10.1029/2024JF007815, 2024.

Merghadi, A., Yunus, A. P., Dou, J., Whiteley, J., ThaiPham, B., Bui, D. T., Avtar, R., and Abderrahmane, B.: Machine learning methods for landslide susceptibility studies: A comparative overview of algorithm performance, Earth Sci. Rev., 207, 103225, https://doi.org/10.1016/j.earscirev.2020.103225, 2020.

Meyrat, G., McArdell, B., Ivanova, K., Müller, C., and Bartelt, P.: A dilatant, two-layer debris flow model validated by flow densitymeasurements at the swiss illgraben test site, Landslides, 19(2), 265–276, https://doi.org/10.1007/s10346-021-01733-2, 2022.

Nguyen, Q.H., Ly, H.B., Ho, L.S., Al-Ansari, N., Le, H.V., Tran, V.Q., Prakash, I. and Pham, B.T.: Influence of data splitting on performance of machine learning models in prediction of shear strength of soil, Math. Prob. Eng., 2021(1), 4832864, https://doi.org/10.1155/2021/4832864, 2021.

Park, S. J., and Lee, D. K.: Predicting susceptibility to landslides under climate change impacts in metropolitan areas of South Korea using machine learning, Geomat. Nat. Hazards Risk, 12(1), 2462-2476, https://doi.org/10.1080/19475705.2021.1963328, 2021.

Park, S., Choi, C., Kim, B., and Kim, J.: Landslide susceptibility mapping using frequency ratio, analytic hierarchy process, logistic regression, and artificial neural network methods at the Inje area, Korea. Environ. Earth Sci. 68(5), 1443-1464,

https://doi.org/10.1007/s12665-012-1842-5, 2013.

Pham, B. T., Prakash, I., Dou, J., Singh, S. K., Trinh, P. T., Tran, H. T., Le, T. M., Phong, T. V., Khoi, D. K., Shirzadi, A., and Bui, D. T.: A novel hybrid approach of landslide susceptibility modelling using rotation forest ensemble and different base classifiers, Geocarto Int., 35(12), 1267-1292, https://doi.org/10.1080/10106049.2018.1559885, 2020.

Riaz, M. T., Basharat, M., Hameed, N., Shafique, M., and Luo, J.: A data-driven approach to landslide-susceptibility mapping in mountainous terrain: case study from the Northwest Himalayas, Pakistan, Nat. Hazards Rev., 19(4), 05018007, https://doi.org/10.1061/(ASCE)NH.1527-6996.0000302, 2018.

Shahabi, H., and Hashim, M.: Landslide susceptibility mapping using GIS-based statistical models and Remote sensing data in tropical environment, Sci. Rep., 5(1), 1-15, https://doi.org/10.1038/srep09899 , 2015.

Shano, L., Raghuvanshi, T. K., and Meten, M.: Landslide susceptibility evaluation and hazard zonation techniques–a review, Geoenviron. Disasters, 7(1), 1-19, https://doi.org/10.1186/s40677-020-00152-0, 2020.

Tang, Y., Feng, F., Guo, Z., Feng, W., Li, Z., Wang, J., Sun, Q., Ma, H. and Li, Y.: Integrating principal component analysis with statistically-based models for analysis of causal factors and landslide susceptibility mapping: A comparative study from the loess plateau area in Shanxi (China), J. Clean. Prod., 277, 124159, https://doi.org/10.1016/j.jclepro.2020.124159, 2020.

Xie, Z., Chen, G., Meng, X., Zhang, Y., Qiao, L., and Tan, L.: A comparative study of landslide susceptibility mapping using weight of evidence, logistic regression and support vector machine and evaluated by SBAS-InSAR monitoring: Zhouqu to Wudu segment in Bailong River Basin, China. Environ. Earth Sci., 76(8), 313, https://doi.org/10.1007/s12665-017-6640-7, 2017.

Xing, Y., Yue, J., Guo, Z., Chen, Y., Hu, J., and Travé, A.: Large-scale landslide susceptibility mapping using an integrated machine learning model: A case study in the Lvliang Mountains of China, Front. Earth Sci., 622,  https://doi.org/10.3389/feart.2021.722491, 2021.

Zhang, Y., Wu, W., Qin, Y., Lin, Z., Zhang, G., Chen, R., et al.: Mapping landslide hazard risk using random forest algorithm in Guixi, Jiangxi, China, ISPRS Int. J. Geo-Inf., 9(11), 695, https://doi.org/10.3390/ijgi9110695, 2020.

Zhu, A. X., Miao, Y., Wang, R., Zhu, T., Deng, Y., Liu, J., Yang, L., Qin, C. Z., and Hong, H.: A comparative study of an expert knowledge-based model and two data-driven models for landslide susceptibility mapping, Catena, 166, 317-327, https://doi.org/ 10.1016/j.catena.2018.04.003, 2018.

Zhou, W., Minnick, M. D., Chen, J., Garrett, J., and Acikalin, E.: GIS-Based Landslide Susceptibility Analyses: Case Studies at Different Scales, Nat. Hazards Rev., 22(3), 05021007, 2021a.

Zhou, X., Wu, W., Qin, Y., and Fu, X.: Geoinformation-based landslide susceptibility mapping in subtropical area, Sci. Rep., 11(1), 1-16, https://doi.org/10.1038/ s41598-021-03743-5, 2021b.